# FrameThinker: Learning to Think with Long Videos via Multi-Turn Frame Spotlighting

**Zefeng He**[12]    **Xiaoye Qu**[1]*    **Yafu Li**[3]    **Siyuan Huang**[4]    **Daizong Liu**[5]    **Yu Cheng**[3]*

[1]Shanghai AI Laboratory    [2]Nanjing University    [3]The Chinese University of Hong Kong
[4]Shanghai Jiao Tong University    [5]Peking University

## Abstract

While Large Vision-Language Models (LVLMs) have achieved substantial progress in video understanding, their application to long video reasoning is hindered by uniform frame sampling and static textual reasoning, which are inefficient and struggle to handle visually intensive video tasks. To overcome these challenges, in this paper, we introduce the concept of thinking with long videos and propose a novel framework FrameThinker. Within this framework, LVLMs are able to iteratively interrogate video content. Developing such video reasoning capabilities in LVLMs presents notable challenges, particularly in adapting the model to new video actions (e.g. select frame), and designing reward functions to guide LVLMs to adopt the newly introduced action. To solve these challenges, we propose a two-phase training strategy, first employing Supervised Fine-Tuning (SFT) to instill fundamental action capabilities, followed by Reinforcement Learning (RL) to optimize a strategic decision-making policy. Notably, in this RL phase, we conduct an in-depth and comprehensive exploration of the reward design for each action and format reward. Extensive experiments on reasoning benchmarks like Video-Holmes, LongVideo-Reason, and long-video understanding benchmarks such as LongVideoBench, MLVU, VideoMME, and LVBench, demonstrate that FrameThinker gets a significant average improvement of +10.4% over baselines while drastically reducing the number of processed frames. Most notably, our 7B model, FrameThinker establishes a new state-of-the-art on LongVideo-Reason, achieving 76.1% accuracy using an average of only 20.6 frames. This not only outperforms the competitive LongVILA-R1 (72.0%) but does so with over 20x fewer frames (vs. 512), demonstrating unparalleled efficiency and effectiveness. Our code is available at: `https://github.com/lcqysl/FrameThinker`.

## 1 Introduction

In recent years, the remarkable advancements of Large Vision-Language Models (LVLMs) have been extended into the challenging domain of video understanding. State-of-the-art closed-source models such as Gemini-2.5 (Comanici et al., 2025), GPT-4o (OpenAI, 2024) and GPT-5 (OpenAI, 2025a), alongside powerful open-source counterparts like Qwen2.5-VL (Bai et al., 2025), have established new performance benchmarks on a suite of video understanding benchmarks, such as Video-MME (Fu et al., 2025a) and LVBench (Wang et al., 2024a).

However, a fundamental limitation still persists in the operational paradigm of existing methods (Bai et al., 2025; Chen et al., 2024c; 2025b). This limitation lies in their reliance on processing a large, pre-selected set of frames obtained through uniform sampling, as illustrated in the top panel of Figure 1. This strategy suffers from inefficiency, especially for long video reasoning, as it processes a large number of irrelevant frames. Meanwhile, such a long and noisy context, cluttered with irrelevant frames, degrades the reasoning performance (Qu et al., 2025a). Furthermore, these methods conduct their reasoning process exclusively through text tokens, which restricts their multimodal perceptual abilities after the initial input. Concurrently, while Video Agents (Fan et al., 2024; Wang et al., 2024b; 2025; Zhang et al., 2025b) can interact with videos using tools, they often depend on

---
* Corresponding authors.

predefined workflows or external models, which limits their autonomy and flexibility in dynamically exploring video content based on reasoning needs. Furthermore, most of these methods are not end-to-end learnable from data, which restricts their potential for improvement.

To address these limitations, we introduce FrameThinker, a novel framework that empowers the model to perform an active and iterative analysis of the video content. As clearly illustrated in the bottom panel of Figure 1, our model first conducts a sparse scan (Turn 1) to establish a broad and general understanding. Guided by this initial assessment, it then identifies promising temporal segments and executes a targeted "zoom-in" by selecting a specific sequence of frames (Turn 2) to retrieve more fine-grained and detailed visual information. This iterative refinement process can be repeatedly applied as needed, fully guided by the model's own reasoning, thereby allowing the model to dynamically gather the necessary evidence while processing only a fraction of the

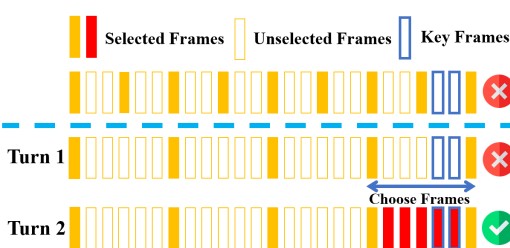

Figure 1: **(Top)** Traditional Uniform Sparse Sampling is inefficient and may miss the key frame in long videos. **(Bottom)** Our method starts with a sparse scan for an overview (Turn 1), then dynamically chooses frames on promising segments (Turn 2), enabling a multi-turn analysis to efficiently focus on key frames in a long video.

total frames. In this way, our FrameThinker constructs a multimodal chain of thought that interleaves textual reasoning with visual frames, ultimately empowering the model to "think with long videos".

Our training methodology facilitates this capability in two stages: we first employ Supervised Fine-Tuning (SFT) to acquaint the model with the core mechanics of action execution (e.g. select frame), followed by Reinforcement Learning (RL) to cultivate an optimal policy for strategic action. Furthermore, we conduct an extensive exploration of the reward function design space for multi-turn video analysis. Among our key findings, we identify that unconditional action rewards are highly prone to mode collapse. While shifting to a simple conditional reward structure mitigates this issue, it does not prevent the model from learning to output illogical actions. To address this deeper challenge, we further propose a Cognitive Consistency Verification (CCV) module to suppress illogical executions and ensure the rationality and interpretability of the model's actions.

To validate the effectiveness of FrameThinker, we evaluate it on complex reasoning benchmarks like Video-Holmes (Cheng et al., 2025) and LongVideo-Reason (Chen et al., 2025b), as well as on long-video comprehension benchmarks including LongVideoBench (Wu et al., 2024b), MLVU (Zhou et al., 2024), VideoMME (Fu et al., 2025a), and LVBench (Wang et al., 2024a). Across all tested benchmarks, our model consistently achieves superior results, outperforming the baseline by an average of 10.4% in accuracy while using significantly fewer frames. Notably, on the LongVideo-Reason benchmark, FrameThinker achieves an accuracy of 76.1% using merely 20.6 frames on average, surpassing the competitive LongVILA-R1 (Chen et al., 2025b) (72.0%) which requires 20 times more frames (512), and establishing a new state-of-the-art. Furthermore, on the reasoning-intensive Video-Holmes benchmark, our method achieves an accuracy of 46.8%, which also establishes a new state-of-the-art while using remarkably fewer frames.

To summarize, our main contributions are threefold:

- We introduce FrameThinker for long video reasoning, a novel framework that empowers LVLMs to actively and dynamically focus on reasoning-intensive video frames, shifting the paradigm from passive video processing to active, multi-turn iterative reasoning.

- We conduct a comprehensive investigation of the reward design space for multi-turn video reasoning. Our deep dive into reward conditioning reveals that even conditional rewards can reinforce illogical thought-action pairs. This motivated our proposal of the Cognitive Consistency Verification (CCV) module to enforce logical consistency and interpretability.

- We demonstrate through extensive experiments that FrameThinker achieves superior accuracy while using significantly fewer frames than previous methods, establishing new state-of-the-art performance on challenging reasoning benchmarks.

## 2 RELATED WORK

**Video Understanding.** LVLMs have significantly advanced video understanding (Liu et al., 2021; 2024a;b; 2020; Qu et al., 2020; Fang et al., 2026; 2024b;a). Following the release of DeepSeek-R1 (Guo et al., 2025), a new wave of models based on Reinforcement Learning with Verifiable Reward (RLVR), including Video-R1 (Feng et al., 2025), VideoChat-R1 (Li et al., 2025), and LongVILA-R1 (Chen et al., 2025b), has demonstrated substantial performance gains, even surpassing proprietary models on certain benchmarks (Chen et al., 2025b). However, these methods predominantly rely on a passive, uniform frame sampling strategy, which limits their efficiency and effectiveness on long videos. In this paper, we challenge this paradigm by introducing Frame-Thinker, a framework that empowers the model to actively and iteratively interrogate the video.

**Video Agents**. Video agents (Wang et al., 2024b; Fan et al., 2024; Wang et al., 2025; Zhang et al., 2025b; Liu et al., 2025) typically rely on external tools to perform complex tasks. With improvements in both these tools and LVLMs, video agents now achieve strong performance on multiple benchmarks. Despite their success, most methods rely on fixed workflows, with decision policies that cannot be learned from data, which limits their flexibility. In contrast, our approach learns an autonomous, end-to-end policy directly from data, enabling the model to decide when and how to interact with the video while leveraging its reasoning to guide visual exploration.

**Thinking with Image.** Traditional LVLMs (Bai et al., 2025; Chen et al., 2024c; 2025b) typically process visual information as a static starting condition, after which all subsequent reasoning unfolds purely in the textual domain. Recently, the emerging paradigm of "Thinking with Images" (Su et al., 2025a;b;c; OpenAI, 2025b; Zheng et al., 2025) has challenged this approach by enabling models to iteratively consult and leverage visual data as an integral part of their thought process, leading to improved performance. Inspired by this evolution, we propose FrameThinker, which empowers the model to dynamically query a video for relevant frames based on its evolving cognitive state, thereby enabling the model to "think with long videos".

## 3 METHOD

### 3.1 FRAMEWORK OVERVIEW

As illustrated in Figure 2, instead of processing the video in a single, passive pass, our FrameThinker framework engages in a multi-turn reasoning loop, actively and strategically seeking information from the video for its decision-making. At each step, the model first generates a textual thought and then selects a specific action from a predefined action space. This explicit separation of reasoning and action is articulated through a structured output format: `<think>...</think><action>...</action>`. Formally, for a given input query $q$, the model produces a sequence of thought-action-observation triplets, namely trajectory $\tau$:

$$\tau = ((t_1, a_1, o_1), \ldots, (t_n, a_n, o_n)),  \tag{1}$$

where for a given query $q$ at step $j$, $t_j$ represents the textual thought (i.e `<think>` tag) generated by the model. This component captures the model's explicit reasoning for its subsequent action. $a_j$ is the action (i.e `<action>` tag) selected by the model. $o_j$ is the observation returned by the environment after executing action $a_j$. For instance, $o_j$ could be a set of frames, a frame number, or a terminal signal. $n$ is the total number of steps in the trajectory $\tau$.

More complex and diverse examples of the model's reasoning process are provided in Appendix D. In this paper, we explore the below three actions for long video reasoning:

- `choose frames between START_FRAME and END_FRAME`: This is the primary action for visual exploration. It allows the model to retrieve a sequence of frames (e.g., 8 frames) from a specific temporal segment of the video, enabling a "zoom-in" capability.
- `get frame number at time MM:SS`: This auxiliary action enhances the model's temporal awareness. It translates a human-readable timestamp into a precise frame index, which can then be used in subsequent `choose frames` actions.
- `output answer`: This is a terminal action that concludes the reasoning process. The model executes this action when it has gathered sufficient evidence to confidently provide the final answer to the user's query.

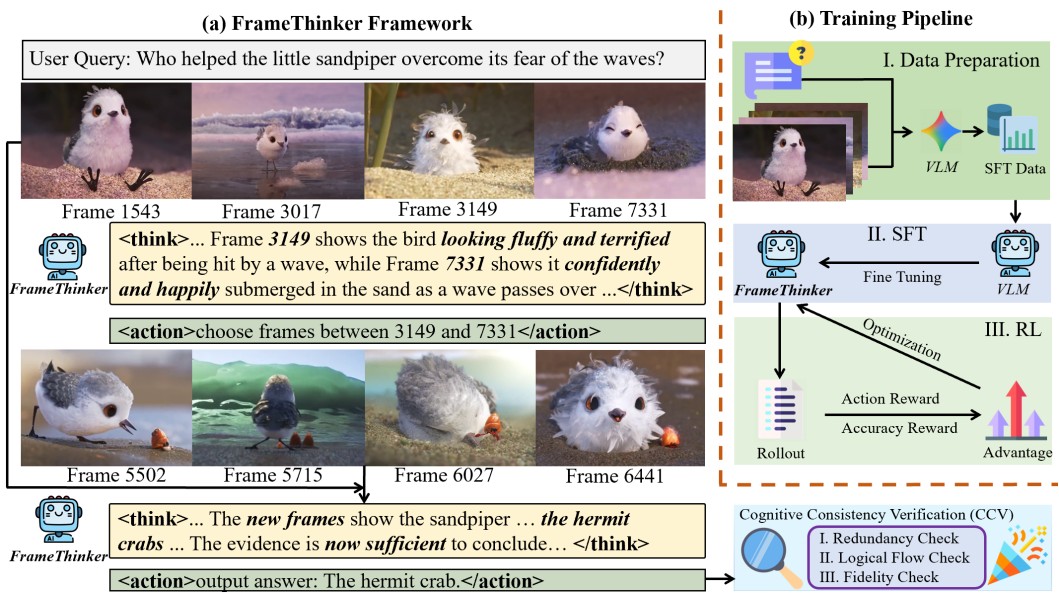

Figure 2: (a) An illustration of the iterative reasoning process of our proposed FrameThiner. The model first performs a sparse scan, then uses thought-action steps to progressively gather evidence. The CCV module ensures this process is logically consistent and interpretable. (b) Our three-stage training pipeline, consisting of Data Preparation, Supervised Fine-Tuning (SFT) to learn action syntax, and Reinforcement Learning (RL) to optimize the policy.

## 3.2 SUPERVISED FINE-TUNING

To equip our FrameThinker model with the correct syntax for the defined actions, we first conduct a Supervised Fine-Tuning (SFT) phase. This phase utilizes a small, curated dataset of only 2,392 examples, designed specifically to instill the basic action grammar. The composition of this data is illustrated in the left panel of Figure 3, with a detailed breakdown provided in Appendix G. The more complex task of learning a strategic policy is deferred to the subsequent Reinforcement Learning (RL) phase.

With this curated SFT dataset, we fine-tune the model using a standard autoregressive language modeling objective. The model is trained to predict the next token in the ground-truth trajectories, which are composed of both <think>

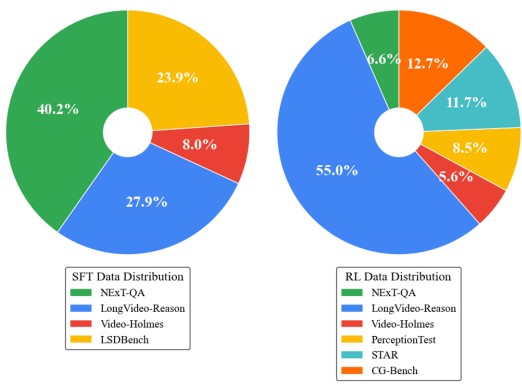

Figure 3: The distribution of data sources for the SFT (Left) and RL training phases (Right).

and <action> sequences. Specifically, we compute a standard cross-entropy loss, but only on the tokens generated by the model (i.e., the content within the <think> and <action> tags). The input query and the observations from action execution are treated as context and are excluded from the loss calculation. This phase ingrains the fundamental mechanics of action execution, preparing the model for the more strategic learning in the subsequent RL phase.

## 3.3 REINFORCEMENT LEARNING (RL)

While the SFT phase successfully equips the model with the basic syntax of action execution, its supervised nature has inherent limitations. Reliance on a small, fixed dataset can create a tendency for the model to memorize specific solution paths, rather than truly generalizing the underlying reasoning strategy (Chu et al., 2025; Zhang et al., 2025a; Qu et al., 2025b; Yan et al., 2025). This approach leads to fragile policies that suffer from limited generalization when faced with novel scenarios. To transition the model from memorization to true generalization, we introduce the Reinforcement

Learning (RL) phase. Crucially, this phase utilizes a much larger and more diverse dataset of 28k examples, which discourages overfitting to specific patterns and compels the model to learn a more robust, generalizable policy. The composition of this dataset is illustrated in the right panel of Figure 3, with further details available in Appendix G.

We adopt Group Relative Policy Optimization (GRPO) (Shao et al., 2024; Guo et al., 2025) to optimize the policy during the RL phase. For each query $q$, we sample $G$ trajectories $\{\tau_i\}_{i=1}^{G}$ from the old policy $\pi_{\theta_{\text{old}}}$ and normalize their outcome rewards to construct the advantages. We omit the KL divergence term to improve efficiency and avoid prematurely constraining the policy's search for an optimum. (Yu et al., 2025). The objective is defined as:

$$\mathcal{J}_{\text{GRPO}}(\theta) = \mathbb{E}_{\tau_i \sim \pi_{\theta_{\text{old}}}} \left[ \frac{1}{G} \sum_{i=1}^{G} \min \left( r_i A_i, \; \text{clip}(r_i, 1 - \epsilon, 1 + \epsilon) A_i \right) \right] \tag{2}$$

$$\text{where} \quad r_i = \frac{\pi_\theta(\tau_i \mid q)}{\pi_{\theta_{\text{old}}}(\tau_i \mid q)}, \qquad A_i = \frac{R_i - \text{mean}(\{R_1, ..., R_G\})}{\text{std}(\{R_1, ..., R_G\}) + \delta} \tag{3}$$

Here, $R_i$ is the reward for trajectory $\tau_i$, $\epsilon$ is the clipping hyperparameter, and $\delta$ is a small constant for numerical stability.

Given that the interactive, multi-turn nature of our framework introduces a unique context for video understanding, it raises new questions for reinforcement learning, particularly regarding how to design an effective reward function. Therefore, in this paper, we make the first attempt to comprehensively explore the design space of the reward function in detail in the following paragraphs.

### 3.3.1 SHOULD WE USE A FORMAT REWARD?

Reinforcement Learning with Verifiable Reward (RLVR) frameworks (Guo et al., 2025) often include a format reward. We initially explored this reward but observed a critical issue: the model quickly learned to suppress action usage, discouraging exploration, as shown in our ablation study (Section 4.3). The cause is that a large format reward (e.g., comparable to accuracy reward) creates a perverse incentive. Early in training, the model discovers that outputting a final answer without reasoning, even a random guess, provides a low-risk way to stably earn the format reward. Attempting actions, in contrast, risks a format error and a zero reward.

Based on this finding, we deliberately omit the format reward. This choice is further justified by our framework's execution engine, which inherently penalizes malformed actions: an invalid action triggers an execution error, prematurely terminating the trajectory and forfeiting any chance of receiving the final accuracy reward.

### 3.3.2 SHOULD ACTION REWARDS BE UNCONDITIONAL OR CONDITIONAL?

We initially experimented with an unconditional action bonus, where the model receives a reward simply for executing an action, regardless of the final outcome. However, we observed that this approach often leads to model collapse. For instance, the model might learn to repeatedly call the same simple action in a loop or even attempt to execute multiple actions within a single turn, with its reasoning collapsing into meaningless, repetitive text. A detailed analysis of these failure modes is provided in Appendix E.1.

Therefore, we shifted to a conditional action bonus, where rewards are granted only when an action is part of a trajectory that leads to a correct final answer ($R_{\text{acc}} = 1$). Our total reward is formulated as a sum of this outcome-based accuracy reward and a process-based action bonus:

$$R_{\text{total}} = R_{\text{acc}} + R_{\text{action}} \tag{4}$$

where the conditional action bonus, $R_{\text{action}}$, is defined as a weighted sum over the actions taken:

$$R_{\text{action}} = \lambda_{\text{cf}} \cdot \mathbb{I}(\texttt{cf} \in \tau) + \lambda_{\text{gfn}} \cdot \mathbb{I}(\texttt{gfn} \in \tau) \tag{5}$$

where $\lambda_{\text{cf}}$ and $\lambda_{\text{gfn}}$ are the reward weights for `choose frames` and `get frame number`.

While this conditional reward structure resolves the issue of mode collapse, we found that a more subtle challenge still remains: the model might learn to execute spurious or illogical actions that,

by chance, are part of a successful trajectory. For instance, the model might select a different frame interval in its action than what it decided in the thought process, or it might request the frame number for a specific timestamp but then choose frames from a completely different interval (examples can be seen in Appendix E.3). This highlights the need for a mechanism to enforce consistency between the model's reasoning and its subsequent actions.

To prevent the reinforcement of such flawed reasoning paths, we introduce the **Cognitive Consistency Verification (CCV)** module. During training, this rule-based module acts as a filter after rollout. It is applied to every trajectory $\tau$ to validate its logical consistency by checking for redundancy, logical flow, and fidelity between thought and action (detailed in Appendix E.3). Any trajectory that fails these checks is immediately terminated. The final reward, $R_{\text{final}}$, is then calculated as:

$$R_{\text{final}} = R_{\text{total}} \cdot V_{\text{CCV}}(\tau) \tag{6}$$

where $V_{\text{CCV}}(\tau)$ is a verification function that returns 1 if the trajectory $\tau$ passes the Cognitive Consistency Verification, and 0 otherwise. This mechanism effectively suppresses the model's attempts to cheat for rewards by assigning a zero reward to illogical trajectories. During the inference phase, the CCV module serves as a runtime safeguard. If an illogical action is detected, the current attempt is terminated, allowing for a retry or a fallback. The CCV module ensures that the generated chain of thought serves as a verifiable trace of the decision-making process, which is crucial for interpretability. It also yields additional performance gains, as detailed in Section 4.3.

### 3.3.3 WHICH ACTION SHOULD THE REWARD PRIORITIZE?

With the reward structure defined, the next design question is how to set the weights, $\lambda_{\text{cf}}$ and $\lambda_{\text{gfn}}$, to encourage the most effective behavior. We deliberately set $\lambda_{\text{gfn}} \gg \lambda_{\text{cf}}$ because supervising the `choose frames` action is difficult; even if an action is logically consistent and part of a successful trajectory, it does not guarantee the action itself was meaningful. The model could have retrieved irrelevant frames and still reached the correct answer using information from the initial query or previous turns. In contrast, for time-specific tasks, the `get frame number` action is objectively correct and highly informative, providing precise, verifiable information that directly supports subsequent decision-making. This, in turn, provides indirect supervision for the `choose frames` action, as the bonus is only awarded if the model subsequently uses the obtained frame number correctly within its selected interval, a condition enforced by the CCV module.

### 3.3.4 SHOULD WE ENCOURAGE THE MODEL TO PERFORM AS MANY TURNS AS POSSIBLE?

In our multi-turn interactive setting, a natural question arises: should we explicitly encourage the model to take more turns? To investigate this, we replace the previous action reward with a newly designed function that directly promotes additional turns:

$$R_{\text{action}} = k \cdot (T - 1) \tag{7}$$

where $T$ is the total number of turns in the trajectory and $k$ is a small positive constant. The subtraction of 1 accounts for the final action to output answer, which is not an analysis step.

We conducted experiments with both unconditional and conditional rewards and observed that, although the model's average number of turns initially increased, training quickly became unstable and subsequently collapsed, with the model's reasoning degrading into meaningless statements. (Detailed training curves and case studies are provided in Appendix E.2). We conclude that this reward setup makes the training unstable: the model, in its pursuit of higher cumulative rewards, learns to prioritize increasing the number of turns over maintaining coherent reasoning and achieving the actual task objective. Therefore, we did not adopt this configuration in our final design.

## 4 EXPERIMENTS

Our empirical evaluation is conducted in two main stages: small-scale and large-scale experiments. Both stages share an identical Supervised Fine-Tuning (SFT) phase, but they differ in the Reinforcement Learning (RL) phase. Due to resource constraints, the small-scale experiments serve as

Table 1: Performance on Video-Holmes. The table breaks down the overall accuracy into seven sub-tasks: Social Reasoning (SR), Intention & Motive Chaining (IMC), Temporal Causal Inference (TCI), Timeline Analysis (TA), Multimodal Hint Reasoning (MHR), Physical Anomaly Reasoning (PAR), and Core Theme Inference (CTI). *denotes model trained on Video-Holmes.

| Model | Frames | SR | IMC | TCI | TA | MHR | PAR | CTI | Overall |
|---|---|---|---|---|---|---|---|---|---|
| *Closed Source Models* | | | | | | | | | |
| GPT-4o (OpenAI, 2024) | 32 | 50.0 | 49.6 | 38.8 | 30.0 | 44.0 | 39.2 | 37.0 | 42.0 |
| Gemini-1.5-Pro (Team et al., 2023) | - | 52.1 | 48.2 | 34.4 | 26.0 | 39.2 | 46.4 | 38.9 | 41.2 |
| Gemini-2.5-Pro (Comanici et al., 2025) | - | 46.6 | 49.3 | 46.9 | 53.0 | 40.1 | 44.3 | 37.4 | 45.0 |
| *Open Source Models* | | | | | | | | | |
| GRPO-CARE (Chen et al., 2025a) | - | 42.8 | 35.1 | 25.6 | 40.5 | 29.2 | 29.9 | 32.6 | 33.5 |
| GRPO-CARE* (Chen et al., 2025a) | - | 46.2 | 44.9 | 31.5 | 49.5 | 39.2 | 37.1 | 37.4 | 40.7 |
| Video-R1 (Feng et al., 2025) | 32 | 48.6 | 41.7 | 28.9 | 34.5 | 31.0 | 33.5 | 35.9 | 36.5 |
| VideoChat-R1 (Li et al., 2025) | 32 | 42.1 | 38.8 | 24.5 | 39.5 | 29.5 | 27.8 | 29.3 | 33.0 |
| Qwen2.5-VL-7B (Bai et al., 2025) | 32 | 38.4 | 34.8 | 17.6 | 30.0 | 27.1 | 18.6 | 25.2 | 27.8 |
| **FrameThinker (Ours)** | **10.2** | **58.9** | **58.7** | **49.1** | **60.0** | **56.0** | **53.6** | **49.1** | **56.1** |

our primary platform for conducting detailed ablation studies. These studies are crucial for validating our experimental setup and examining key aspects of the training methodology, such as the design of the reward function. Specifically, for the small-scale experiments, we perform RL training exclusively on the training set of Video-Holmes (Cheng et al., 2025). We choose this benchmark because its emphasis on active clue-seeking for reasoning aligns perfectly with the philosophy of our framework. Conversely, the large-scale experiments are focused on training our final model on our complete RL dataset to achieve the best possible performance and enhance its ability to generalize across diverse problem distributions. The resulting model is then used to report our final results against competing methods on these benchmarks, validating the effectiveness of our approach.

## 4.1 BENCHMARKS AND BASELINES

We evaluate our method on a comprehensive suite of six benchmarks: Video-Holmes (Cheng et al., 2025), LongVideo-Reason (Chen et al., 2025b), LongVideoBench (Wu et al., 2024b), MLVU (Zhou et al., 2024), VideoMME-Long(w/o sub) (Fu et al., 2025a), and LVBench (Wang et al., 2024a). These benchmarks are strategically chosen to assess distinct capabilities. The first two benchmarks, Video-Holmes and LongVideo-Reason, are designed to evaluate the model's advanced reasoning abilities. The subsequent four focus on long-video comprehension across a spectrum of increasing durations, among which VideoMME-Long and LVBench specifically challenge performance in exceptionally long scenarios. Further details on each benchmark are provided in Appendix G.1.

Our baseline models are chosen to provide a comprehensive comparison against leading models. We select Qwen2.5-VL-7B (Bai et al., 2025) as our primary baseline for direct comparison, and we also include Video-R1 (Feng et al., 2025) and VideoChat-R1 (Li et al., 2025), as they are also based on Qwen2.5-VL-7B. Furthermore, for broader context, we also report the performance of other models as cited on the official benchmark leaderboards.

## 4.2 IMPLEMENTATION DETAILS

Our implementation is based on the Qwen2.5-VL-7B-Instruct model. In Reinforcement Learning (RL), the number of frames sampled in each turn was adaptively set: 8 frames for most videos, and 12 frames for longer videos ($> 300\,\text{s}$) to better handle the extended temporal context. The primary reward is based on the final accuracy ($R_{\text{acc}} = 1$), supplemented by action reward, with no format reward. The small-scale experiments used a learning rate of 1.0e-6 and reward bonuses $\lambda_{\text{gfn}} = 0.2$, $\lambda_{\text{cf}} = 0$, while the large-scale training used a learning rate of 5.0e-7 and reward bonuses $\lambda_{\text{gfn}} = 0.5$, $\lambda_{\text{cf}} = 0.02$. Further implementation details are provided in Section B.

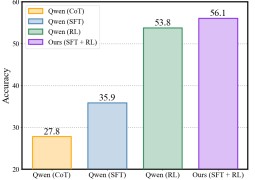 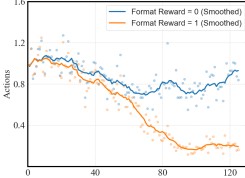 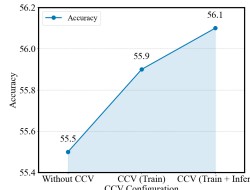 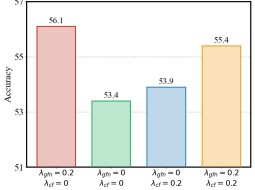

(a) Ablation on the effect of training stages. (b) Ablation on the effect of format rewards. (c) Ablation on the effect of the CCV module. (d) Ablation on the effect of reward configurations.

Figure 4: Comprehensive ablation studies on key components of our training methodology. (a) Direct comparison against a fine-tuned Qwen2.5-VL-7B baseline. (b) Impact of including a format reward. (c) Ablation on the CCV module during training and inference. (d) Performance under different action reward configurations.

### 4.3 SMALL-SCALE EXPERIMENTS

**Main Results.** When trained on Video-Holmes, our model achieves an accuracy of 56.1%, establishing a new state-of-the-art, as shown in Table 1. Our method not only surpasses existing models in accuracy but also use significantly fewer frames on average.

**Ablation Studies.** We conducted a series of ablation studies to quantify the advantages of our framework, with the results visualized in Figure 4.

First, we conducted a direct comparison by fine-tuning the baseline Qwen2.5-VL-7B model using our identical SFT or RL setups. The results, presented in Table 4a, demonstrate that our full approach achieves a clear performance advantage, confirming the efficacy of FrameThinker's architecture.

Next, we present the impact of format reward. As shown in Figure 4b, the version trained with a format reward exhibits a rapid decline in actions during the early stages of training, indicating that such a reward can inadvertently suppress the model's incentive to explore and utilize actions.

Furthermore, we validated the effectiveness of our Cognitive Consistency Verification (CCV) module. The results in Figure 4c show its impact when applied during the training and inference phases, underscoring that the CCV module's contribution extends beyond just enhancing interpretability; it also leads to a tangible improvement in performance.

Finally, we investigated the impact of different configurations for action reward, with the results presented in Figure 4d. Our findings indicate that assigning a larger reward bonus to the `get frame number` action than to the `choose frames` action (in this case, $\lambda_{\text{gfn}} = 0.2$ and $\lambda_{\text{cf}} = 0$) yields the most significant performance benefit. We attribute this to the more accurate and reliable information provided by the `get frame number` action.

### 4.4 LARGE-SCALE EXPERIMENTS

In our large-scale experiments, we conducted a comprehensive evaluation of FrameThinker across six challenging benchmarks, demonstrating its consistent superiority in both accuracy and frame efficiency. The model's strength is particularly evident on benchmarks designed to test complex reasoning abilities (Table 2). For instance, on LongVideo-Reason, our model establishes a new state-of-the-art result. It achieves 76.1% accuracy using merely 20.6 frames on average, surpassing the strong LongVILA-R1 baseline which requires 512 frames.

Table 2: Performance on reasoning benchmarks. †indicates results evaluated by us. *denotes model trained on Video-Holmes.

| Model | Video-Holmes | | LongVideo-Reason | |
|---|---|---|---|---|
| | Frame | Acc | Frame | Acc |
| *Closed Source Models* | | | | |
| GPT-4o (OpenAI, 2024) | 32 | 42.0 | - | - |
| Gemini-1.5-Pro (Team et al., 2023) | 32 | 41.2 | - | 69.3 |
| *Open Source Models* | | | | |
| LongVILA (Chen et al., 2024c) | - | - | - | 62.7 |
| LongVILA-R1 (Chen et al., 2025b) | - | - | 512 | 72.0 |
| GRPO-CARE (Chen et al., 2025a) | - | 33.5 | - | - |
| GRPO-CARE* (Chen et al., 2025a) | - | 40.7 | - | - |
| Video-R1 (Feng et al., 2025) | 32 | 36.5 | - | 68.1 |
| VideoChat-R1 (Li et al., 2025) | 32 | 33.0 | 32 | 67.2† |
| Qwen2.5-VL-7B (Bai et al., 2025) | 32 | 27.8 | 32 | 64.1† |
| **FrameThinker (Ours)** | 15.9 | **46.8** | 20.6 | **76.1** |
| Δ | -50% | +19.0 | -36% | +12.0 |

This trend of high performance and exceptional frame efficiency continues across the four long-video comprehension benchmarks (Table 3). When viewed holistically, FrameThinker's well-rounded capabilities become clear. The radar chart in Fig-

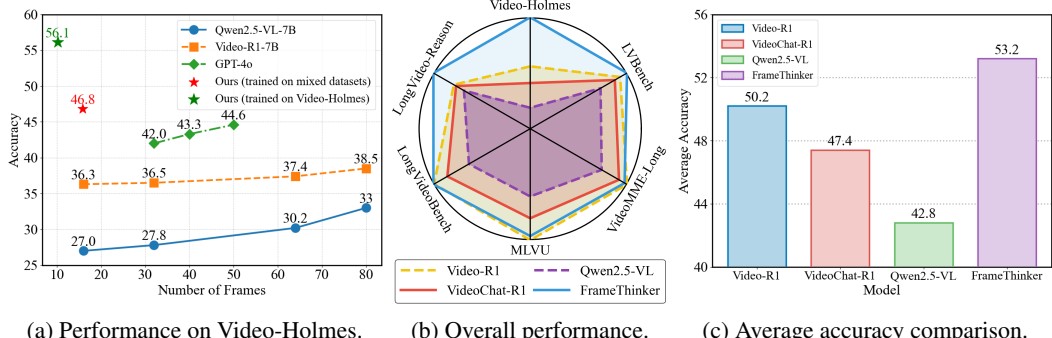

(a) Performance on Video-Holmes.   (b) Overall performance.   (c) Average accuracy comparison.

Figure 5: (a) Accuracy and the number of frames processed on Video-Holmes. (b) A radar chart comparing overall performance across six benchmarks, where results are normalized and scaled for visual comparison. (c) Average accuracy across six benchmarks. Our FrameThinker achieves the best average performance.

Table 3: Performance on long-video benchmarks. †indicates results evaluated by us.

| Model | LongVideoBench | | MLVU | | VideoMME-Long | | LVBench | |
|---|---|---|---|---|---|---|---|---|
| | Frame | Acc | Frame | Acc | Frame | Acc | Frame | Acc |
| *Closed Source Models* | | | | | | | | |
| GPT-4o (OpenAI, 2024) | 32 | 58.5 | 0.5fps | 64.6 | 384 | 65.3 | 60 | 48.9 |
| Gemini-1.5-Pro (Team et al., 2023) | 32 | 55.2 | - | - | 0.5fps | 67.4 | 3600 | 33.1 |
| *Open Source Models* | | | | | | | | |
| PLLaVA (Xu et al., 2024) | 16 | 40.2 | - | - | - | - | 16 | 26.1 |
| ShareGPT4Video (Chen et al., 2024b) | 16 | 39.7 | 16 | 46.4 | 16 | 35.0 | - | - |
| LongVA (Zhang et al., 2024) | - | - | 256 | 56.3 | 128 | 46.2 | - | - |
| VITA-1.5-7B (Fu et al., 2025b) | - | - | - | - | 128 | 46.2 | - | - |
| Video-R1 (Feng et al., 2025) | 32 | 52.7† | 32 | 60.2† | 32 | 48.2† | 32 | 35.3† |
| VideoChat-R1 (Li et al., 2025) | 32 | 49.1† | 32 | 54.3† | 32 | 46.2† | 32 | 34.3† |
| Qwen2.5-VL-7B (Bai et al., 2025) | 32 | 43.2† | 32 | 48.4† | 32 | 41.9† | 32 | 31.6† |
| **FrameThinker (Ours)** | 21.1 | **52.9** | 23.2 | 59.1 | 24.1 | 47.6 | 23.9 | **36.6** |
| Δ | -34% | +9.7 | -28% | +10.7 | -25% | +5.7 | -25% | +5.0 |

ure 5b provides a visual summary, illustrating its advantage over competing methods in various benchmarks. Quantitatively, this superiority is best captured by its average performance, as shown in Figure 5c. FrameThinker achieves an average accuracy of 53.2%, representing a significant improvement of +10.4% over the baseline Qwen2.5-VL-7B. (Qwen2.5-VL-7B and its derivatives were evaluated using a Chain-of-Thought prompting strategy, with the specific prompt detailed in Appendix C.3). For detailed training curves, see Appendix F.

## 4.5 COMPUTATIONAL EFFICIENCY ANALYSIS

While FrameThinker demonstrates unparalleled efficiency in terms of the number of frames processed per task, we further evaluate its real-world performance by analyzing the Wall-Clock Latency (Time). The results are summarized in Table 4.

The results in Table 4 clearly validates that FrameThinker's frame-efficient, active reasoning strategy translates directly into superior real-world computational speed. FrameThinker

Table 4: Computational Efficiency and Wall-Clock Latency on LongVideo-Reason. Inference was conducted with HuggingFace Transformers.

| Model | Frame | Acc | Time (s) |
|---|---|---|---|
| Video-R1 | 32 | 68.1 | 8.6 |
| LongVILA-R1 | 256 | 70.8 | 38.1 |
| LongVILA-R1 | 512 | 72.0 | OOM |
| **FrameThinker** | **20.6** | **76.1** | **7.7** |

achieves a new state-of-the-art accuracy of 76.1% on LongVideo-Reason while simultaneously maintaining an extremely low average latency of just 7.7 seconds per task.

## 5  CONCLUSION

In this paper, we introduced FrameThinker, a novel framework that advances video understanding by shifting the paradigm from passive, uniform sampling to active, iterative analysis. Our model is trained to dynamically interrogate video content, intelligently selecting and reasoning over a minimal number of frames to find the answer. Our training pipeline consists of Supervised Fine-Tuning (SFT) and Reinforcement Learning (RL). Within this framework, we further explore the reward design space for multi-turn video analysis, and introduce a Cognitive Consistency Verification (CCV) module to ensure that the model's actions remain logically grounded, interpretable, and aligned with its reasoning. Our extensive experiments show that FrameThinker achieves state-of-the-art performance on challenging reasoning benchmarks while drastically reducing the required visual context. This work demonstrates a significant advance in video understanding, creating powerful models that can achieve more comprehensive reasoning with drastically less visual context.

## ETHICS STATEMENT

Our research is grounded in the use of publicly available, established academic benchmarks for video understanding. We have not collected any new data containing personal or sensitive information. The primary goal of our work is to advance the efficiency and reasoning capabilities of LVLMs, which we believe is a positive contribution to the research community. However, we acknowledge that like any powerful video analysis technology, FrameThinker could potentially be misused for applications such as large-scale surveillance or automated content filtering, which could have negative societal impacts. Our intention is strictly for research purposes, and we did not design the framework for such applications. By open-sourcing our code and models, we aim to foster transparency and encourage further research into both the capabilities and the potential societal implications of such models, allowing the community to collaboratively develop appropriate safeguards.

## REPRODUCIBILITY STATEMENT

To ensure the reproducibility of our results, we will make our code and model checkpoints publicly available upon publication. The core code is currently provided for review via an anonymous link. Our work is implemented based on the open-source Qwen2.5-VL-7B-Instruct model. All datasets used for the Supervised Fine-Tuning (SFT), Reinforcement Learning (RL), and final evaluation phases are publicly available academic benchmarks. We provide a detailed breakdown of these datasets in Appendix G. Our key experimental configurations are thoroughly described in the Implementation Details (Section 4.2). Furthermore, the specific prompts used for our model and the baseline evaluations are provided in Appendix C. We believe these resources provide sufficient detail for the community to verify our findings and build upon our work.

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

APPENDIX

APPENDIX CONTENTS

## A  USE OF LLMS

During the preparation of this manuscript, we utilized LLM to aid in the writing process. The LLM's role was limited to polishing the text and improving its readability. Specifically, it was used for proofreading to identify and correct grammatical errors, spelling mistakes, and awkward phrasing. The authors reviewed and edited all LLM-generated suggestions to ensure accuracy and take full responsibility for the final content of the paper.

## B  MORE IMPLEMENTATION DETAILS

**Additional Training Details.** In the original setup, Supervised Fine-Tuning (SFT) was performed using parameter-efficient LoRA (Hu et al., 2022) with a rank of 8 on 8 H800 GPUs. A per-device batch size of 1 with 4 gradient accumulation steps was used, resulting in an effective batch size of 32. The initial learning rate was set to 1.0e-4, and a cosine learning rate scheduler with a warmup ratio of 0.1 was applied. Reinforcement Learning (RL) training was conducted on 8 H800 GPUs with a batch size of 32 and a rollout number of 8.

## C    PROMPTS

### C.1    SYSTEM PROMPT FOR FRAMETHINKER

The detailed system prompt used to guide FrameThinker during our experiments is shown in Figure 6. This prompt defines its available actions, and the required format for its thought and action process.

> You are an expert AI assistant that answers questions about a video by iteratively analyzing it.
> Your task is to output your reasoning within a <think> </think> tag, followed by a specific action within an <action> </action> tag.
> Possible actions are:
> 1. `choose frames between START_FRAME and END_FRAME`: Request a more detailed view of a specific video segment. The number of frames is fixed, currently {CONFIG["NUM_FRAMES_TO_SAMPLE"]}.
> 2. `get frame number at time MM:SS`: Get the exact frame number for a specific time. Convert hours to minutes if needed (e.g., for 1 hour, 2 minutes, and 30 seconds, use 62:30).
> 3. `output answer: OPTION`: Provide the final answer (e.g., A, B, C...) when you are confident.

Figure 6: The system prompt for FrameThinker. It outlines its core task, the specific syntax for action usage (`get frame number` and `choose frames`), and the structured `<think>...<action>` output format.

### C.2    FALLBACK PROMPT FOR CCV FAILURES

During the inference phase, the CCV module acts as a runtime safeguard. If it detects an illogical action (i.e., a failure in the Logical Flow or Fidelity Check), the ongoing iterative process is terminated. To prevent the model from getting stuck or failing the entire task, a fallback mechanism is triggered. The model is then presented with a new, simplified system prompt, as shown in Figure 7. This prompt instructs the model to provide a direct, final answer directly, which ensures a robust response.

> You are an expert AI assistant that answers questions about a video.
> Your task is to output your reasoning within a <think> </think> tag, followed by a specific action within an <action> </action> tag. Your only action is: `output answer: OPTION`: Provide the final answer (e.g., A, B, C...).

Figure 7: The fallback system prompt triggered upon a CCV failure during inference. It directs the model to formulate a final answer directly.

### C.3    CHAIN-OF-THOUGHT PROMPT FOR BASELINES

Additionally, we provide the Chain-of-Thought (CoT) prompt used for evaluating baseline models, such as Qwen2.5-VL-7B and its derivatives, illustrated in Figure 8. To ensure a fair comparison, we adopted the same evaluation prompt as used in the original Video-Holmes benchmark (Cheng et al., 2025).

> Based on the given video, reason and answer the single-choice question. Provide your reasoning between the <think> and </think> tags, and then give your final answer between the <answer> and </answer> tags. The question is: {question}. The options are: {options}. Your answer:

Figure 8: The Chain-of-Thought prompt used for evaluating baseline models.

## D    ADDITIONAL EXAMPLES

To further illustrate the capabilities of our framework, we present additional qualitative examples. In the following cases, only a small subset of the frames retrieved and processed is visualized.

Figure 9 demonstrates how the model handles a precise, timestamp-specific query by first using an auxiliary action to locate the event.

Question: What does the disappearance of wall reflection when pressing the switch at 0:13 imply?
A. Switch control gas valve
B. There is a malfunction in the electrical circuit inside the house.
C. The switch is in the off position when it is turned upwards.
D. The angle of specular reflection changes
E. Someone cut off the power supply.
F. Produce the phenomenon of visual persistence

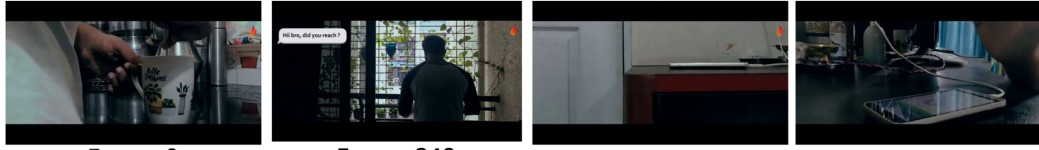

Frame 0                    Frame 849

**\<think>**The user is asking about an event that happens at a specific timestamp, 0:13. To understand what this event implies, I first need to locate the exact frame at this timestamp.**\</think>\<action>**get frame number at time 00:13**\</action>**

Frame number at time 00:13 is: 311.

**\<think>**The frame number is 311, which falls between the initially provided frames **0** and **849**. Observing the frames between frame **0** and frame **849** should provide the necessary detail.**\</think>\<action>**choose frames between 0 and 849**\</action>**

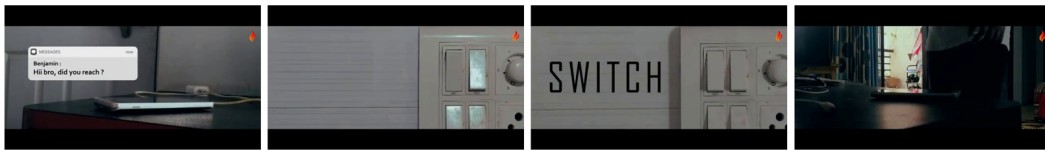

**\<think>**The new set of frames confirms that the switch is being turned on and off. The reflection on the wall disappears when the switch is in the off position. This directly supports option C. The disappearance of the reflection is a clear indicator of the switch's state. I have enough information to make a decision.**\</think>\<action>**output answer: C**\</action>**

Figure 9: An example of our framework solving a reasoning task. Given a question about a specific timestamp (0:13), the model first formulates a thought and executes the `get frame number` action. Based on the returned frame number, it initiates a second reasoning turn, using the `choose frames` action to "zoom in" on the relevant scene. This multi-turn, iterative process allows the model to gather the necessary visual evidence (the state of the light switch's reflection) to arrive at the correct answer.

Figure 10 highlights the model's ability to solve ambiguous, non-timestamp-specific questions through its iterative exploration strategy.

Question: What is the relationship between the man in black and the man in red?
A. brother
B. Hunter and Fugitive
C. friend
D. Doctors and patients
E. colleague
F. stranger

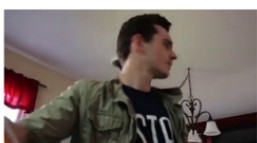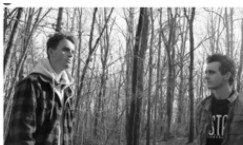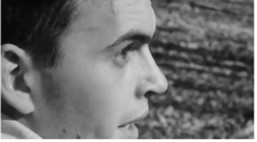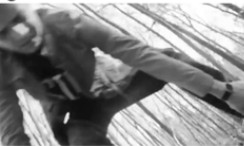

Frame 2259                                    Frame 3766

<think>The user wants to understand the relationship between the two men. The initial frames show a man in a green jacket and another in a red hoodie. The scene transitions from peaceful to a confrontation in the woods, which suggests a conflict. The most dramatic and revealing moment seems to be around frame **3766**, where a physical struggle is depicted. To understand their relationship better, I need to examine the sequence of events more closely between the initial meeting and the fight. I'll focus my search on the segment between frame **2259** and frame **3766**. </think><action>choose frames between 2259 and 3766</action>

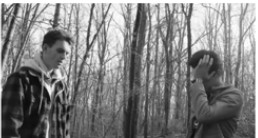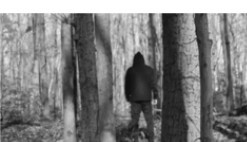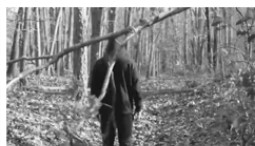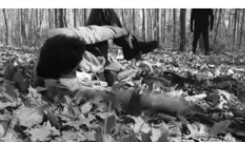

<think>The new set of frames confirms my hypothesis. The man in black is shown as the aggressor, chasing and eventually attacking the man in red. This dynamic is best described as a \"Hunter and Fugitive\". The man in red is clearly fleeing and being pursued. The evidence is now overwhelming. </think><action>output answer: B</action>

Figure 10: An example of resolving an ambiguous query without a specific timestamp. Given only sparse initial frames that suggest a conflict, the model's first thought is to hypothesize a crucial temporal segment between the peaceful beginning and the depicted struggle. It then executes a `choose frames` action to "zoom in" on this interval. The newly retrieved, denser frames provide unequivocal evidence of a chase and attack, allowing the model to confidently deduce the "Hunter and Fugitive" relationship and select the correct answer. This case demonstrates the effectiveness of the model's iterative refinement process for complex narrative understanding.

Figure 11 presents a more complex reasoning trajectory, showcasing the model's ability to perform a hierarchical, multi-step search.

Question: What does the girl use to find the person she is looking for in the video?
A. Mask
B. Tail
C. Phone
D. Car

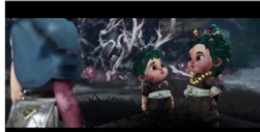 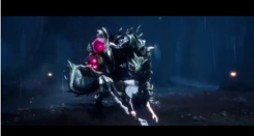 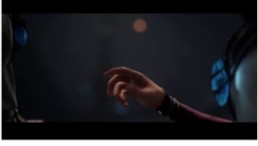 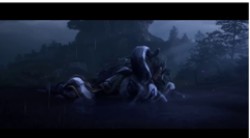

Frame 16407          Frame 32814

<think>The user wants to identify the tool used by the girl to find someone. Looking at the initial frames, we see two children (frame **16407**) and then a large creature (frame **32814**). The question likely pertains to an event that happens between these two points. To find the object, I need to examine the segment more closely. I'll focus my search between frames **16407** and **32814**.</think><action>choose frames between 16407 and 32814</action>

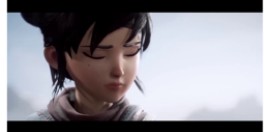 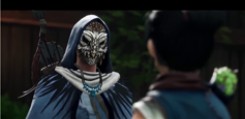 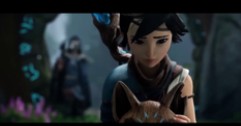 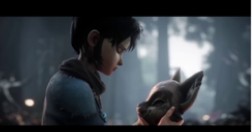

Frame 23438          Frame 25782          Frame 28125

<think>The new set of frames clearly shows a woman in a hood (frame **23438**) giving a mask to the girl (frame **25782**, **28125**). This action seems to be directly related to the girl finding the person she is looking for. I have seen the mask being given and then being worn. The key event is between frames **25782** and **28125**. I need to examine this part more closely to confirm that this mask is used to find the person. I will focus my search between frames **25782** and **28125**.</think><action>choose frames between **25782** and **28125**</action>

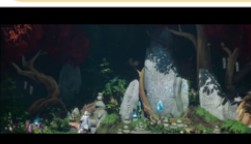 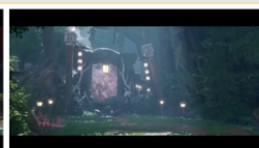 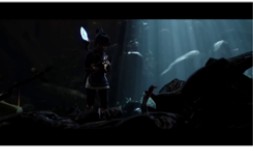 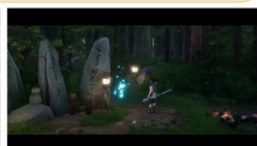

<think>The new frames confirm that the girl wears the mask and uses it to navigate through the forest, finding the way to the final battle. The mask is clearly shown being given to her and then being worn by her. I have gathered enough information. The correct option is A, Mask. I am confident in my answer.</think><action>output answer: A</action>

Figure 11: An example of a multi-step reasoning process that showcases hierarchical search. The model begins with a broad search (`choose frames between 16407 and 32814`) based on the sparse initial frames. After this first action reveals the mask as a key object, the model formulates a new thought and performs a second, more targeted search (`choose frames between 25782 and 28125`) to confirm its function. This iterative refinement strategy, moving from a coarse to a fine-grained analysis, demonstrates the model's advanced capability to solve complex problems by efficiently focusing on the most relevant visual evidence.

Figure 12 demonstrates a case where the reasoning process reconsiders the initial analysis and performs a targeted search on a completely new temporal segment.

Question: How did the camera wearer prepare and spread ingredients on the dough before adding toppings?
A. Spread chopped onions and sprinkled pepper on the dough.
B. Spread butter and garlic on the dough before adding herbs.
C. Layered tomato slices and grated cheese on the dough.
D. Arranged potato slices and chopped onions on the dough, then sprinkled salt.

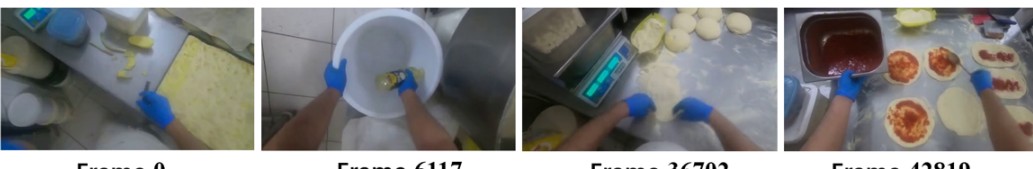

**Frame 0**      **Frame 6117**      **Frame 36702**      **Frame 42819**

<think>... Frame **42819** shows toppings being added. Frame **36702** shows the dough being prepared. It's logical to assume that the ingredient spreading step mentioned in the question occurs between the dough preparation and the sauce application. Therefore, I should investigate the segment between frame **36702** and frame **42819** to see how the ingredients are applied to the dough before the sauce. I will choose frames between **36702** and **42819**.</think><action>choose frames between **36702** and **42819**</action>

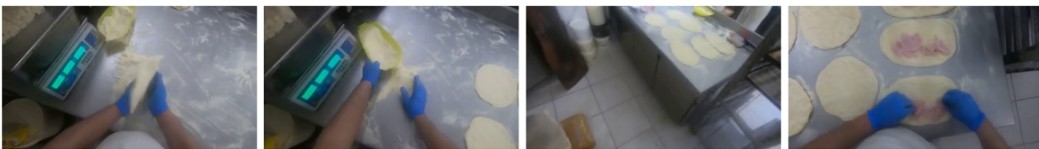

<think>My previous analysis led me to a section ... This did not align with any of the options provided. ... Let me reconsider the initial set of frames. Frame **0** is particularly interesting. It shows a large, rectangular piece of dough covered with a layer of what looks like sliced potatoes. A person is also seen slicing potatoes on the counter. This scene is quite different from the later frames showing the preparation of round pizzas. It's plausible that the question refers to this specific preparation at the beginning of the video, which I may have initially overlooked. The question asks how ingredients were prepared and spread before toppings. The process in frame **0** seems to be exactly that. To verify this, I need to see what happens immediately after frame **0**. ... I will choose frames between **0** and **6117**.</think><action>choose frames between **0** and **6117**</action>

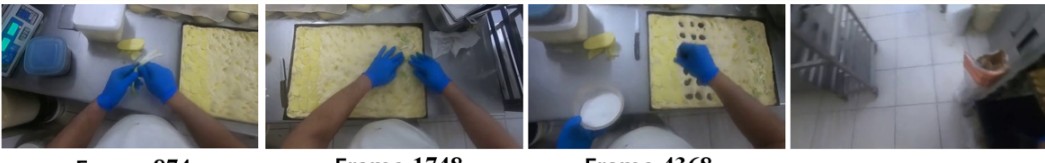

**Frame 874**      **Frame 1748**      **Frame 4368**

<think>The analysis of the frames from the beginning of the video has provided a clear sequence of actions. The initial frames show the person slicing potatoes and arranging them on a rectangular dough. Subsequent frames show the preparation and spreading of chopped onions (or a similar vegetable like leeks) on another section of the dough (frames **874** and **1748**). Frame **4368** clearly depicts the person sprinkling salt from a bowl over the prepared dough. This sequence of arranging potato slices and chopped onions, followed by sprinkling salt, directly corresponds to option D. Therefore, I have sufficient evidence to confidently select the correct answer.</think><action>output answer: D</action>

Figure 12: An example of a complex reasoning trajectory featuring Reconsideration and Multi-interval Exploration. Based on a sparse initial scan, the process initially focuses on a segment between Frame 36702 and 42819 (identified as dough and sauce preparation). After the first targeted search fails to provide a matching option, the trajectory identifies a key visual difference in Frame 0 (showing rectangular potato-covered dough) compared to later frames (showing round pizzas). The reasoning then reconsiders the initial segment and shifts focus to a new, completely different temporal segment (`choose frames between 0 and 6117`), thereby showcasing a flexible multi-interval exploration strategy. The new set of frames provides the necessary visual sequence (arranging potato slices, chopped onions, and sprinkling salt), which leads to a confident selection of the correct answer (D). This case highlights the robustness of the policy in self-correction.

In contrast to the multi-turn examples that require exploration, Figure 13 illustrates a scenario of optimal efficiency where the model performs a "Direct Answer."

Question: What appears on the screen at the very beginning of the video?
A. The lake
B. The mountain with green trees
C. The prairie
D. The snow mountain

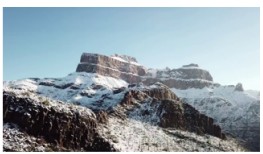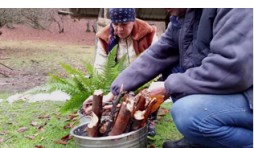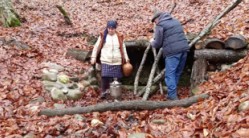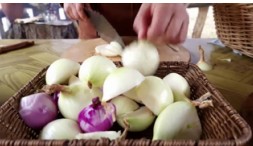

**Frame 0**

**<think>**The user is asking what appears on the screen at the very beginning of the video. The initial frame provided, frame **0**, clearly shows a snow-covered mountain. This directly corresponds to option D. The subsequent frames show different scenes, but the beginning is unambiguously the mountain. I am confident in this answer based on the first frame alone. **</think><action>**output answer: D**</action>**

Figure 13: An example of Direct Answering. The user asks about the very beginning of the video. The model correctly identifies that the provided Frame 0 already contains the necessary information. In its thought process, it recognizes the sufficiency of the initial evidence and confidently proceeds directly to the `output answer` action without invoking any actions. This case demonstrates the model's strategic capability to not only explore when necessary but also to conclude efficiently when the answer is readily available, avoiding redundant actions.

# E    BAD CASE

## E.1    MODE COLLAPSE FROM UNCONDITIONAL REWARDS

In our initial explorations of reward design for reinforcement learning, we observed that improper reward mechanisms could lead the model to learn invalid, repetitive strategies—a phenomenon known as mode collapse. This issue was particularly pronounced when providing unconditional rewards for specific actions, as the model would learn to exploit the reward function rather than solve the task.

**Unconditional reward for the `get frame number` action.** We first experimented with a reward function where a fixed, positive bonus, specifically set to 0.2, was granted if the model executed the `get frame number` action at least once within a trajectory, regardless of the final answer's correctness. This unconditional incentive led to a severe mode collapse, as illustrated in Figure 14. The model's behavior degraded in two significant ways. First, its reasoning process became incoherent and circular; the content within the `<think>` tags devolved into nonsensical, repetitive phrases like "I need to first" without any substantive analysis of the video content. Second, the model learned to repeatedly execute the exact same action (`get frame number at time 00:22`) without any intention of using the resulting information, demonstrating a clear failure to engage in meaningful, goal-oriented exploration.

The underlying cause of this collapse is that the model discovered a simple, low-effort strategy to maximize its reward. The unconditional action bonus provided a high-certainty, immediate reward, whereas the reward for a correct final answer ($R_{acc}$) was uncertain and required complex, multi-step reasoning. Consequently, the model learned a degenerate policy: ignore the task and simply perform the rewarded action to get the bonus. This strategy effectively circumvents the intended learning objective, discouraging any genuine attempt to understand the video and solve the problem. This critical observation underscored the necessity of making action rewards conditional on the final task success, a principle we adopted in our final reward design.

**Unconditional reward for the `choose frames` action.** Similarly, we observed a distinct but related form of mode collapse when providing an unconditional reward of 0.2 solely for the `choose frames` action. In this scenario, the model is rewarded for executing this action, again, without any

Question: What does the scene of balloons bursting one by one at 0:22 imply?
A. Atmospheric pressure changes
B. Birthday magic takes effect.
C. Electrical short-circuit sparks
D. The crowd sabotaged in secret.
E. Invisible creatures move
F. Time loop sign

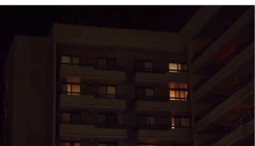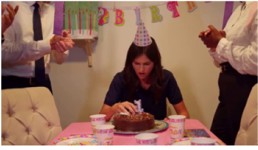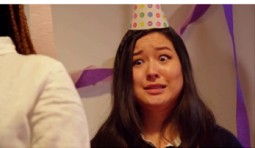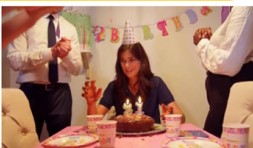

<think>The question asks about a specific event described in the question stem.
First, I need to verify the premise of the question.
I first need to confirm the question's premise.
I need to first determine the question's context.
The question asks about "balloons bursting".
To get context on this question, I'll first need to find information about this event.
I need to first establish if this question is asking about a movie plot.
I need to first understand the question.
I need to find the question's question.
I will first investigate the question's question.</think><action>get frame number at time 00:22</action>

Frame number at time 00:22 is: 527.

<think>After getting the frame number and choosing the question, I need to understand the question's question.
First, I need to assess the question's premise.
The question likely refers to a horror movie plot.
I need to analyze the question to get context.</think><action>get frame number at time 00:22</action>

Frame number at time 00:22 is: 527.

<think>After getting the frame number, I need to understand the question's question.</think><action>get frame number at time 00:22</action>

Frame number at time 00:22 is: 527.

<think>After getting the frame number, I need to investigate the question.
I need to analyze the question.</think><action>output answer: E</action>

Figure 14: An example of mode collapse induced by unconditional reward for the `get frame number` action.

dependency on the final outcome. As shown in Figure 15, the model's response degenerates into a single, elongated turn. Within this turn, the model's reasoning is again replaced by nonsensical, tautological statements about "options and choices." Crucially, it proceeds to execute the `choose frames` action multiple times in quick succession. These actions are often repetitive or illogical, lacking any strategic motivation derived from a coherent thought process.

The cause of this behavior is analogous to the previous case. The model identifies that repeatedly outputting the rewarded action is an efficient strategy for accumulating rewards. Instead of engaging in a multi-turn dialogue to incrementally gather evidence, it learns to "stuff" a single turn with as many rewarded actions as possible before concluding with a likely random guess. This instance of mode collapse further reinforces our conclusion that unconditional rewards are fundamentally flawed for training a strategic model. They incentivize the model to master the exploitation of the reward function itself, rather than to learn the complex, causal reasoning required to solve the actual task. This led us to develop a conditional reward system, which we detail in the main paper.

### E.2 MODE COLLAPSE FROM MULTI-TURN REWARDS

**Reward for More Turns.** Also, we investigated an alternative reward designed to encourage more extensive reasoning. This was implemented using the step-based reward function, shown in Equation 7. For this experiment, we set the reward for each turn to 0.2 (corresponding to $k = 0.2$ in the

Question: What is the most likely relationship between Emma and the dentist?
A. accomplice
B. husband and wife
C. doctor-patient
D. Stranger
E. relatives
F. colleague

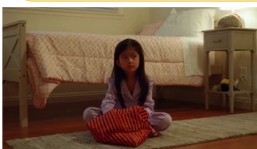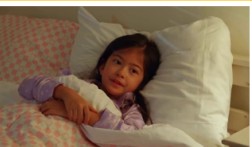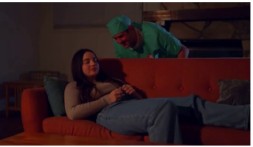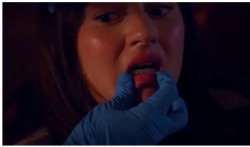

**\<think\>**The user is asking about options and choices that likely appear in the options or choices' descriptions or choices' options list.\</think\>\<action\>get frame number at time 1:30\</action\>\<think\>The user is asking about choices or options related to choices or options about choices or options about options.\</think\>\<action\>choose frames between 1768 and 1856\</action\>\<think\>This action describes options and choices that likely discuss options or choices about choices' options or choices' options.\</think\>\<action\>choose frames between 1768 and 1856\</action\>\<think\>This action explains the choices available in the options list.\</think\>\<action\>choose frames between 1768 and 2656\</action\>\<think\>This option discusses choices and options related to options about choices or options.\</think\>\<action\>choose frames between 1768 and 2656\</action\>\<think\>This option explains the context and choices presented in the choices.\</think\>\<action\>choose frames between 1768 and 2656\</action\>\<think\>This action explains the choices and options for this specific question's question options.\</think\>\<action\>choose frames between 1768 and 2656\</action\>\<think\>This option explains the options or choices' options and their options**.\</think\>\<action\>**output answer: C**\</action\>**

Figure 15: An example of mode collapse induced by unconditional reward for the `choose frames` action.

equation), with the total bonus capped at 0.6. We explored both an unconditional and a conditional version of this reward.

First, in the unconditional setting where the bonus is granted regardless of the final outcome, the training process proved highly unstable, as illustrated in the top row of Figure 16. Initially, the strategy appears successful: the average number of interaction turns per trajectory steadily increases, peaking at over three. However, around step 270, the training abruptly collapses, with the average number of turns plummeting to a degenerate state. This policy collapse is mirrored in the average response length, which simultaneously destabilizes and drops. As shown in Figure 17, the model's thought process is entirely eliminated, with the `<think>` tag simply mirroring the `<action>`. It then terminates the process in the subsequent turn with a random answer, completely failing the task.

In addition, we investigated a conditional version, where the bonus was granted only upon a correct final answer. However, we observed a similar pattern of training instability, as shown in the bottom row of Figure 16. Although the policy collapse occurred at a different point in training (around step 170), the overall dynamic was similar: the average number of turns initially rose and then catastrophically dropped. This demonstrates that directly incentivizing the model to take more turns, even when the reward is conditional, remains an unstable objective that leads to policy collapse.

The reason for this collapse is rooted in the inherent instability of multi-turn reinforcement learning, which this reward design drastically exacerbates. Training a policy for long-horizon, multi-turn interactions is intrinsically challenging due to issues like high variance and difficult credit assignment. The model must learn to maintain a coherent reasoning state across many steps. The reward function in Equation 7, however, creates a perverse incentive that steers the model away from this difficult task. As the graphs vividly show, the model learns that the path of least resistance is not to solve the problem, but to prioritize extending the trajectory. It abandons the complex goal of coherent reasoning in favor of a simpler policy: generate just enough valid syntax to prompt the next turn. This leads to the observed collapse, forcefully demonstrating that the quality and purpose of interactions, rather than their sheer quantity, are what must be rewarded.

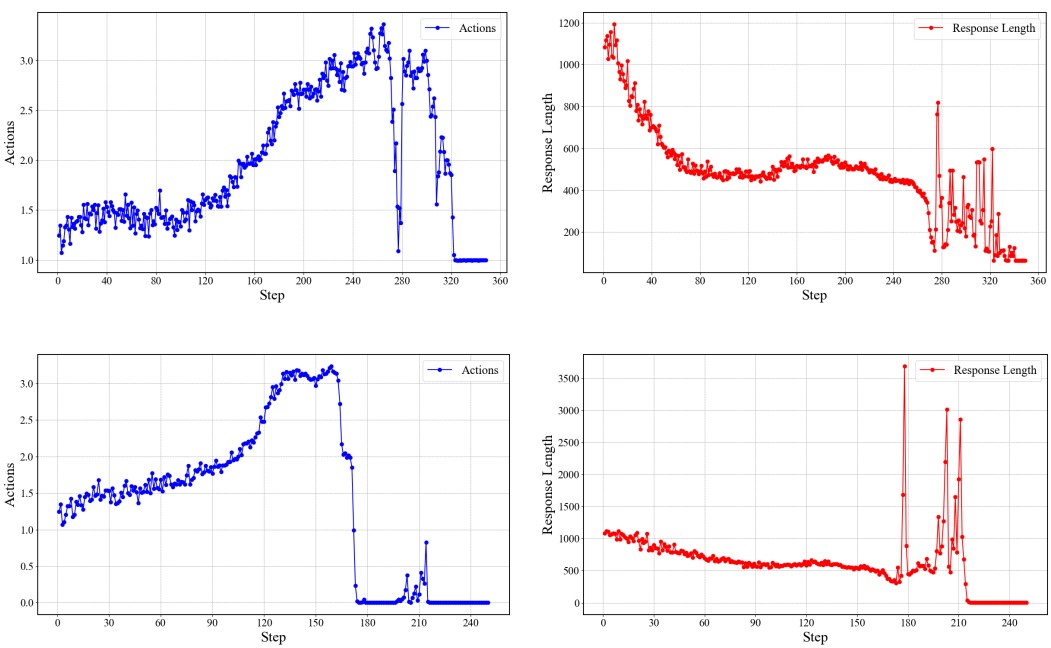

Figure 16: Training dynamics. **(Top Row)** Metrics from training with an unconditional reward. **(Bottom Row)** Metrics from training with a conditional reward. The final `output answer` action is not included in the action counts.

Question: What is the hint of the girl's phone note shown in the 0:15 shot?
A. The other party's number is encrypted.
B. Dialing the emergency call.
C. The remark shows "Danger".
D. The contact person is familiar with the object.
E. The phone battery is about to run out.
F. The address book is empty.

**\<think\>**choose frames between 0 and 6000**\</think\>\<action\>**choose frames between 0 and 6000**\</action\>**

**\<think\>**output answer: D**\</think\>\<action\>**output answer: D**\</action\>**

Figure 17: An example of mode collapse induced by a reward that unconditionally encourages a higher number of turns.

### E.3    TRAJECTORIES FAILING COGNITIVE CONSISTENCY VERIFICATION (CCV)

The Cognitive Consistency Verification (CCV) module is designed to filter out and penalize trajectories that exhibit illogical or inconsistent reasoning patterns. This ensures that the model is rewarded for coherent reasoning rather than for finding loopholes in the reward system, and enhances the model's interpretability. Below, we detail the three primary failure conditions that the module checks for.

**Failure to Pass the Redundancy Check.** First, we posit that a coherent reasoning process should be progressive and avoid redundancy. Therefore, the CCV module checks if the model executes actions with identical content. Specifically, it flags any instance where an action with the exact same parameters is executed more than once within a single trajectory. For example, repeatedly calling `get frame number at time 00:22` to get the same frame number, or repeatedly executing `choose frames between 100 and 200` to retrieve the same set of frames, would be penalized. Such repetitions indicate that the model is either stuck in a logical loop or is not effectively using the information from its actions to advance its reasoning. We observe that this type of redundancy is primarily present during the early stages of training or in cases of mode collapse, as exemplified in Figure 14 where the model repeatedly requests the same timestamp. Such behavior becomes rare once the training process stabilizes.

**Failure to Pass the Logical Flow Check.** Second, the CCV module enforces a logical sequence for actions that are causally linked. This rule primarily addresses how the model utilizes information from the `get frame number` action. When this action retrieves a specific frame number, subsequent `choose frames` action is expected to make meaningful use of this new information. A failure to do so is flagged as logical inconsistency, as it demonstrates the model is not properly connecting its reasoning steps.

An example of this logical breakdown is presented in Figure 18. The model correctly identifies the need to investigate a timestamp (`0:34`) and successfully executes `get frame number`, which returns frame `815`. This is a critical piece of information obtained through an explicit action. However, in its very next step, the model requests frames from the interval `565-645`. This interval does not contain frame `815`, rendering the previous action useless and indicating a disjointed reasoning process. The CCV module identifies this failure to utilize causally linked information and terminates the trajectory.

Question: What was Jeremy's intention in supporting the back of the bundle at 0:34?
A. Prevent items from slipping off.
B. Adjust the bundling angle
C. Relieve lower back pain
D. Cover the outline of the human body
E. Check the tightness of the bundle
F. Soothe the emotions of the bound.

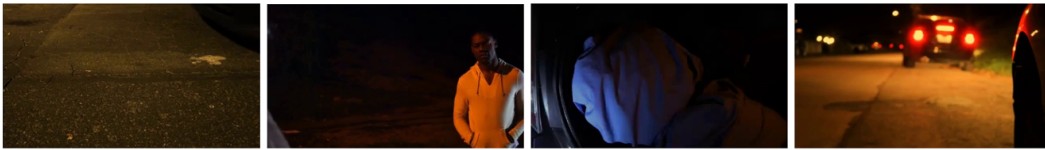

<think>The user is asking about an event that happens very early in the video, at 0:34. First, I need to find the exact frame number to determine the context around this specific action.</think><action>get frame number at time 0:34</action>

Frame number at time 0:34 is: 815.

<think>The user\'s question is about an action that likely occurs near frame 815. The action mentioned in the question "supporting the back of the bundle at 0:34" is probably located around frame 815. The options related to this specific event are likely located in the frames around this frame number. The action described in the question is probably located in the initial frames I\'ve been provided.</think><action>choose frames between 565 and 645</action>

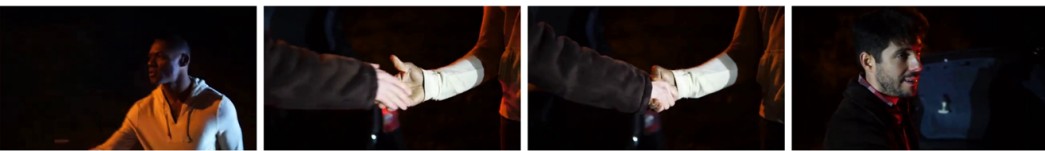

<think>Upon examining the first frame and the action of grabbing a hand, it\'s clear that the "bundle" mentioned in the question is not a physical object but a person. The action at frame 815 (0:34) likely refers to a person being tied up. The action described in the question is probably located in frame 632. The action about supporting the back is not a physical action related to preventing slipping (ruling out A). There\'s no evidence for adjusting the angle (ruling out B). There\'s no mention of pain (ruling out C). There\'s no evidence for checking tightness (ruling out E). Option F seems plausible as it\'s a general action in the context. The term "bundle" is used metaphorically. The action related to this event is likely described in frame 610-620. The context around this action is probably located in frame 610, which is at frame 815 in the question.</think><action>output answer: D</action>

Figure 18: An example of a Logical Flow Check failure. The model first successfully retrieves frame `815` corresponding to a specific timestamp. However, its subsequent `choose frames` action targets an interval (`565-645`) that does not contain the retrieved frame, breaking the logical chain of actions.

**Failure to Pass the Fidelity Check.** Finally, the CCV module ensures that the model's executed action is faithful to its preceding thought process. The module parses numerical arguments from both the `<think>` tag and the subsequent `<action>` tag to check for contradictions. This check is crucial for preventing the model from learning to generate plausible-sounding reasoning that is detached from its actual behavior, thereby suppressing the reinforcement of flawed or "hallucinated" reasoning paths.

Figure 19 presents a clear example of a trajectory that fails this fidelity check. In the `<think>` block, the model's reasoning identifies that the key event is located near frame `4974`. However, the executed action, `choose frames between 1400 and 1500`, completely disregards this analysis and targets an entirely unrelated video segment. By identifying this logical inconsistency, the CCV module terminates the trajectory and assigns a zero reward, preventing such flawed reasoning-action pairs from being positively reinforced.

Question: What does the cheesy flying saucer design at the end imply?
A. Human dependence on dairy products
B. The earth's environment is polluted.
C. The girl's hallucination is visualized.
D. Government secret experiment device
E. Deformation of space debris
F. Alien creatures' trapping strategies

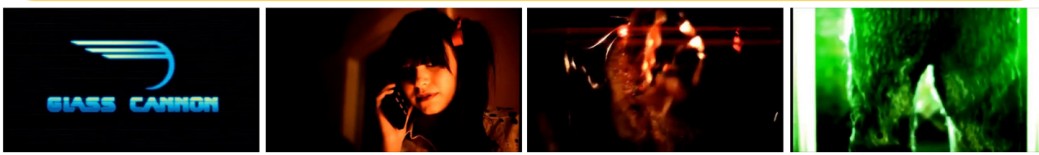

**\<think\>**The user is asking about an event that likely happens towards the end of the video. The question mentions a "cheesy flying saucer design," which is probably located near frame 4974. The term "cheesy" is often used to describe a plot hole or a silly explanation in a movie. The action must be located near frame 4974.**\</think\>\<action\>**choose frames between 1400 and 1500**\</action\>**

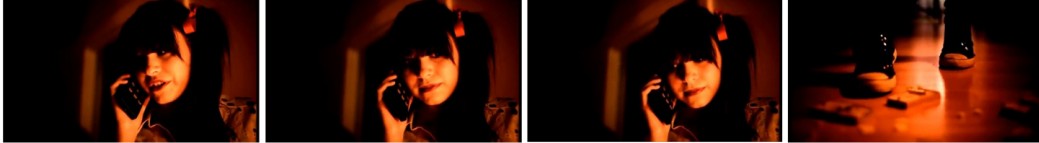

**\<think\>**Upon closer inspection of frame 1499, it\'s clear the frames show a supernatural or horror element rather than a literal "cheesy" explanation. This suggests the option about the "cheesy flying saucer design" is likely near frame 1499. This frame range shows a supernatural theme. The question about the "cheesy" design\'s implication must be located in this range. The term "cheesy" in this context is probably used to describe a plot point. This specific event is located in frame 1499.**\</think\>\<action\>**output answer: C**\</action\>**

Figure 19: An example of a Fidelity Check failure. The model's reasoning in the `<think>` block identifies the relevant area around frame 4974, but the executed `<action>` targets an entirely unrelated interval (1400-1500). The CCV module flags this trajectory as inconsistent.

### E.4   PREMATURE STOPPING

In this section, we present an example of a FrameThinker limitation: Premature Stopping. This failure mode is often characterized by missed events due to poor initial scans or an overly aggressive attempt to conclude the reasoning process. The case is illustrated in Figure 20.

For extremely long videos, sparse sampled frames are inherently limited and may not provide sufficient information to pinpoint the exact location of the answer. As shown in the example, the model correctly analyzes the query and identifies the beginning of the video as the relevant area for the "opening caption" and correctly executes the `choose frames` action on the initial segment (frames 0 to 7997). However, even with this targeted scan, the model still fails to secure the explicit visual information (the target year) in the newly sampled frames. The model prematurely gives up on further visual exploration. Instead, it attempts to make an unmature inference based solely on

the incomplete historical context of the quote it has found. This speculative deduction leads to an incorrect final answer (A) and highlights a challenge in policy learning where the model prioritizes early termination over comprehensive evidence gathering.

Question: What year appears in the opening caption of the video?
A. 1636
B. 1366
C. 1363
D. 1633

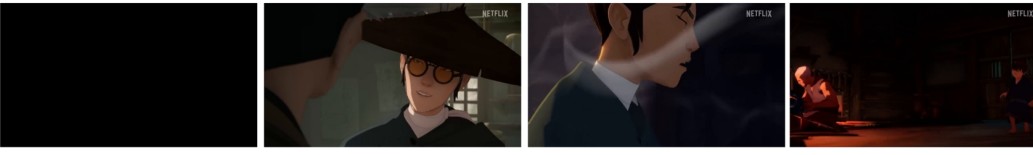

Frame 0        Frame 7997

**\<think\>**The user is asking for the year shown in the opening caption of the video. Based on the initial frames, the video starts at frame **0** and the year is likely present in this initial segment. To find the specific year, I need a closer look at the beginning of the video. I will examine the frames between frame **0** and frame **7997** to get a clearer picture of the opening credits or caption. Therefore, I will choose to observe the frames between frame **0** and frame **7997**. **\</think\>\<action\>**choose frames between 0 and 7997**\</action\>**

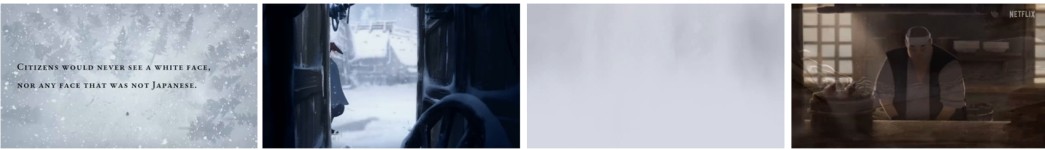

**\<think\>**After reviewing the frames from 0 to 7997, I can see a text card with a quote. While the year isn't explicitly written on this card, it's part of the opening sequence. The options are 1636, 1366, 1363, and 1633. The historical context of the quote suggests a period in Japan's history. The year 1636 is famous for the Edo period's strict isolationist policies, which aligns with the quote "Citizens would never see a white face, nor any face that was not Japanese." The other years are too early for this specific historical context. Based on this evidence, the opening caption refers to the year 1636. I am now confident in this conclusion. **\</think\>\<action\>**output answer: A**\</action\>**

Figure 20: An example of a Premature Stopping failure. The model selects the correct temporal segment but fails to find the target year, opting instead to guess based on an incomplete, historical inference. The correct answer is D.

## F    TRAINING DYNAMICS

The dynamics of the large-scale Reinforcement Learning (RL) training phase are illustrated in Figure 21. These metrics provide a comprehensive overview of how the model's policy and behavior evolve as it learns to interact with the video environment to solve tasks.

As shown in Figure 21a, the primary objective—average accuracy—exhibits a consistent upward trend, demonstrating that the policy is successfully learning. The learning curve begins to converge around step 1100 as the model's performance on the task distribution approaches its peak.

The average action reward (Figure 21b) displays a more complex, two-phase pattern. It rises initially and then stabilizes, before resuming a significant upward trajectory after approximately 850 steps, a point which corresponds to roughly one full epoch over the training data. We interpret this as the model shifting its optimization focus as it begins its second pass through the dataset. In the first epoch (up to step 850), the model learns the fundamental use of actions to solve the more straightforward cases, establishing a solid baseline accuracy. Once this primary accuracy reward begins to plateau, the policy must refine its strategy to earn rewards on more challenging problems. As it re-encounters the data in the second epoch, it learns to employ actions with greater precision and skill. This mastery of more nuanced action use is necessary to solve these harder instances,

resulting in a higher average action reward and allowing the model to continue increasing its total reward even as accuracy gains become marginal.

Figure 21c reveals the evolution of the model's action utilization strategy. The training begins with a noticeable drop in the average number of actions. We attribute this to the model entering the RL phase with an incomplete mastery of the action space, as it was trained on only a small SFT dataset and had not yet fully learned the correct format and usage of actions. The RL process quickly penalizes erroneous or superfluous actions, leading to this sharp corrective decline. Subsequently, as the model accurately learns the utility of beneficial actions, their usage rate recovers and fluctuates before settling into a stable range, indicating convergence to a more purposeful and efficient policy.

Finally, the average response length (Figure 21d) shows a general downward trend. Our SFT data is synthesized using Gemini-2.5-pro (Comanici et al., 2025), a powerful, closed-source model that tends to generate longer, more detailed chains of thought, especially in such multi-turn scenarios. The Qwen2.5-VL-7B base model initially mimics this verbose style. However, during the RL phase, without an explicit reward for length, the policy naturally gravitates towards shorter, more concise outputs. This reflects an optimization towards efficiency, where the model retains the core reasoning necessary for task success while shedding the stylistic verbosity of the teacher model, a behavior more aligned with its own inherent capabilities.

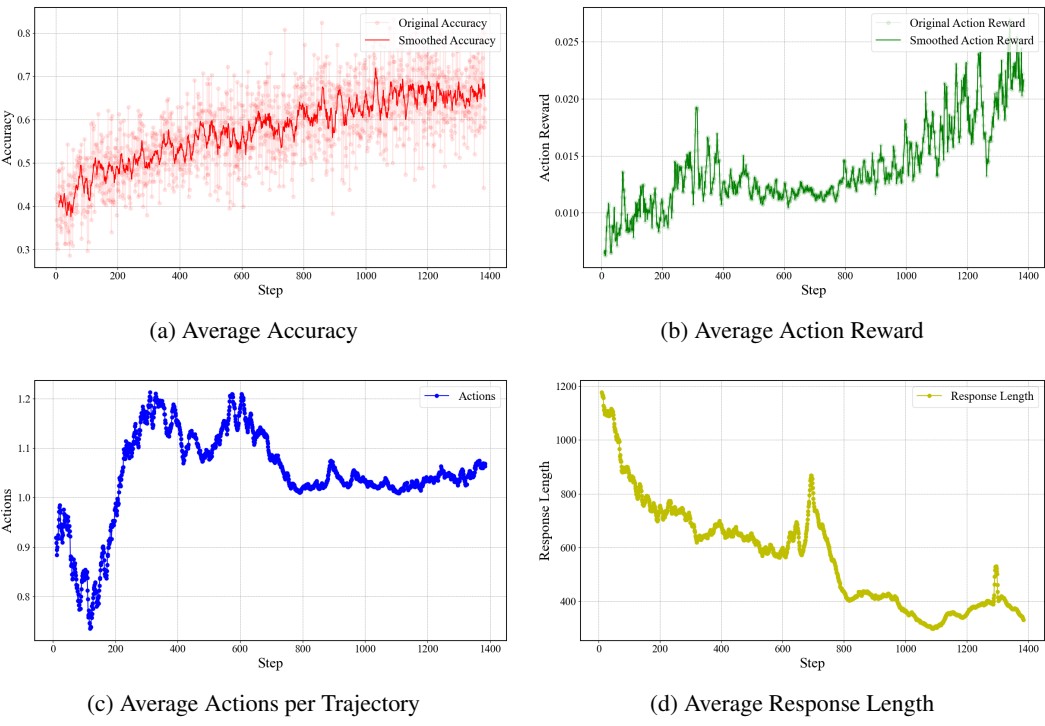

(a) Average Accuracy

(b) Average Action Reward

(c) Average Actions per Trajectory

(d) Average Response Length

Figure 21: Key metrics from the Reinforcement Learning training phase. (a) The model's accuracy on tasks steadily improves. (b) The reward attributed to taking actions increases, showing more effective action use. (c) The number of actions (excluding the output answer) per task stabilizes after an initial exploration phase. (d) The model's generated response (thought + action) becomes more concise over time.

# G  DATASETS AND BENCHMARKS

## G.1  DETAILS OF DATASETS AND BENCHMARKS USED

**Video-Holmes** (Cheng et al., 2025) features short suspense films with rich plots and implicit cues. Its core characteristic is requiring the model to act like a detective, actively searching for and connecting multiple scattered visual and audio clues across different video segments. This process

facilitates deep reasoning about causality, motivation, and themes, far exceeding the simple identification of isolated clues found in traditional benchmarks. We incorporate its training set into our training data and evaluate our final model on the official test set.

**LongVideo-Reason** (Chen et al., 2025b) is designed to comprehensively enhance and evaluate the deep reasoning capabilities of vision-language models on long videos. It includes question-answering content across various complex reasoning types, such as temporal, object-related, spatial, and plot-based inquiries. Similarly, we utilize its training set for our training process and conduct our evaluation on the test set.

**LSDBench** (Qu et al., 2025a) is characterized by its high "Necessary Sampling Density" (NSD) questions. These questions specifically target very brief yet action-dense segments within long videos (e.g., one hour). This design rigorously tests a model's ability to locate and parse critical information amidst a backdrop of redundant data.

**CG-Bench** (Chen et al., 2024a) is a novel, clue-driven question-answering benchmark for long video understanding. Its main feature is the emphasis on requiring the model to answer questions based on specific clues provided within the video context.

**NExT-QA** (Xiao et al., 2021) pushes video understanding from simple action description to deep action explanation. It focuses on reasoning about the causal relationships (why something happened) and temporal relationships (what happened before/after an event) between actions in videos.

**STAR** (Wu et al., 2024a) is a benchmark designed to evaluate a machine's capability for situated reasoning in real-world videos. A key feature is its abstraction of complex dynamic scenes into structured hypergraph representations. It programmatically generates questions centered on four reasoning types—interaction, sequence, prediction, and feasibility—to conduct an in-depth diagnosis of a model's visual perception, contextual understanding, and logical reasoning skills.

**PerceptionTest** (Patraucean et al., 2023) is a novel multimodal video benchmark created to diagnostically evaluate a model's perceptual and reasoning abilities. The dataset contains videos specifically recorded to showcase interesting perceptual scenarios.

**LongVideoBench** (Wu et al., 2024b) is a novel benchmark specifically designed to evaluate the long-context video understanding capabilities of Large Multimodal Models (LMMs). Its core feature is the inclusion of a diverse range of videos with durations of up to one hour.

**MLVU** (Zhou et al., 2024) is a benchmark for evaluating the long video understanding capabilities of Multimodal Large Language Models (MLLMs). It is distinguished by its significant extension of video lengths, ranging from 3 minutes to 2 hours, and its inclusion of diverse video genres such as movies, surveillance footage, and gameplay.

**VideoMME** (Fu et al., 2025a) is the first comprehensive evaluation benchmark for Multimodal Large Language Models in video analysis. Its key characteristic is its all-encompassing design, which covers a wide variety of video types (6 domains, 30 sub-categories) and a broad temporal span, including short, medium, and long videos ranging from 11 seconds to 1 hour.

**LVBench** (Wang et al., 2024a) is characterized by its extremely long video durations, averaging over one hour, which significantly surpasses previous benchmarks of its kind.

We primarily evaluate our method on a comprehensive suite of six benchmarks: Video-Holmes, LongVideo-Reason, LongVideoBench, MLVU, VideoMME-Long (w/o sub), and LVBench. As illustrated in Figure 22, these benchmarks are strategically chosen not only to assess distinct capabilities but also to cover a progressively increasing range of video durations. The first two, Video-Holmes and LongVideo-Reason, are specifically selected to evaluate the model's advanced reasoning abilities on complex narratives. Among them, VideoMME-Long and LVBench specifically test the model on exceptionally long video scenarios, evaluating the robustness of our approach.

### G.2 DETAILS OF SFT DATA

The Supervised Fine-Tuning (SFT) dataset consists of two subsets, as detailed below.

**Process-Supervised Data.** This subset is constructed from five reasoning templates:

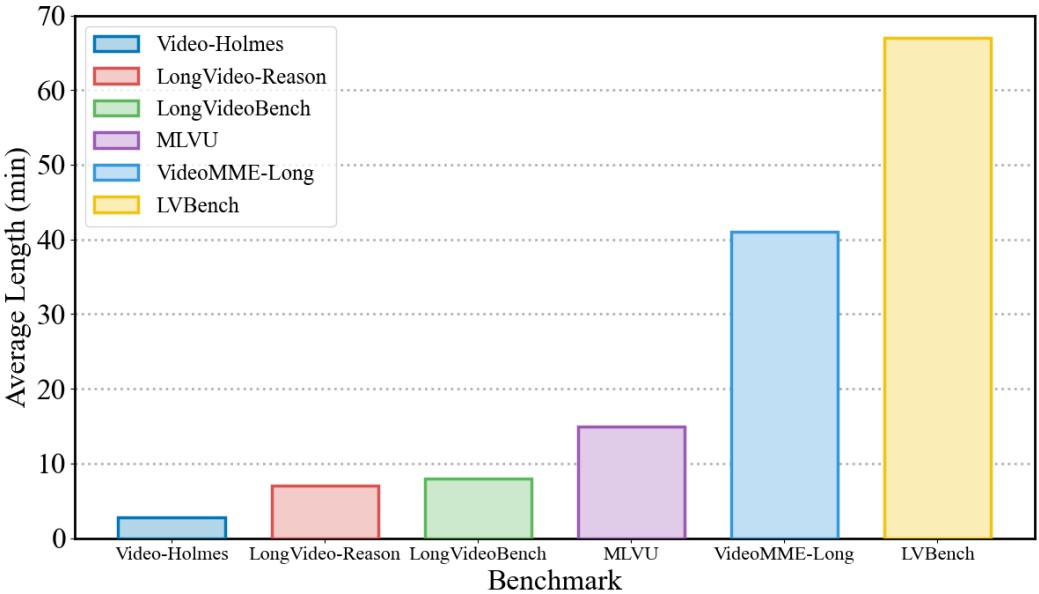

Figure 22: Average video length (in minutes) across the six primary evaluation benchmarks.

A. *Direct Answering:* Answers the query directly without any action.

B. *Single-Pass Frame Selection:* Performs a single `choose frames` action before answering.

C. *Timestamp-Grounded Selection:* Uses `get frame number` for a timestamp, then `choose frames` to inspect the relevant segment.

D. *Hierarchical Frame Selection:* Performs a second `choose frames` action that refines a previously selected interval.

E. *Multi-Interval Exploration:* Performs `choose frames` on two distinct intervals.

**Outcome-Supervised Data.** For this subset, supervision is applied only to the correctness of the final answer, not the reasoning path. The distribution of these reasoning templates, alongside the number of actions per trajectory, is illustrated in Figure 23.

The trajectories for both subsets are synthesized using Gemini-2.5-Pro (Comanici et al., 2025). To ensure a high-quality dataset, we first retain only trajectories that lead to the correct answer and filter out instances with illogical reasoning paths using Gemini-2.5-Flash (Comanici et al., 2025). This curation process ensures that the SFT phase focuses on learning meaningful and effective action execution strategies.

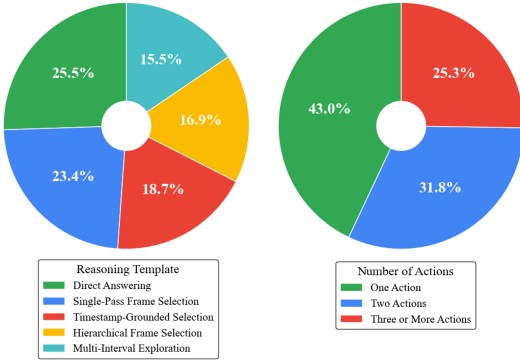

Figure 23: Distribution of reasoning templates (Left) and actions per trajectory (Right).

