# OpenReview forum: "FrameThinker: Learning to Think with Long Videos via Multi-Turn Frame Spotlighting"
_ICLR.cc/2026/Conference — ICLR 2026 Poster_

### Official Review · Reviewer_Qf4G · 2025-10-23

**Soundness:** 3
**Presentation:** 3
**Contribution:** 2
**Rating:** 4
**Confidence:** 3

**Summary:**

Uniform frame sampling and static textual reasoning are the key problems of inefficiency and ineffectiveness in long video reasoning. This paper, FrameThinker, proposes a two stage training approach which consists of a supervised fine-tuning stage to enable the model to make actions (i.e, selecting frames), and a reinforcement learning stage to learn a policy for action decision making. Experiments show that FrameThinker greatly improve the performance over the baseline with much less frames processed, demonstrating the strong effectiveness and efficiency.

**Strengths:**

1. Strong performance over baselines presented in the paper, not only in accuracy, but also efficiency. Evaluations on diverse benchmarks show strong generalizations in the video reasoning.
2. The two stage training, supervised fine-tuning and reinforcement learning, is typical applied in training test-time reasoning in large language models for math and coding problems. This paper expanded the domain to video understanding and reasoning.
3. For reward design, a novel Cognitive Consistency Verification (CCV) module to verify that the actions from the model are logically grounded, interpretable and aligned with its reasoning. Ablation studies show that CCV is crucial to the performance.
3. The figures are well-designed and the paper is easy to follow.

**Weaknesses:**

Missing video agent baseline: there were already some papers adopted a similar high-level idea of selecting video segments/frames for long video reasoning in a coarse-to-fine manner [1]. Therefore, adding at least one video agent baseline can make the contribution stronger if showing superior performance over existing methods.

References:
[1] Yang, et al. Video Curious Agent for Long Video Understanding. 2024.

**Questions:**

1. Benchmarks in the paper are all long-video understanding and reasoning. Will training the model (i.e., FrameThinker) negatively impact the short-video reasoning, and traditional tasks for vision language models?

---

> ### Author Response · Authors · 2025-11-20
> **Response to Reviewer Qf4G (part1)**
>
> We sincerely thank the reviewer for recognizing our strong performance in both accuracy and efficiency across diverse benchmarks, the successful expansion of the two-stage training paradigm to video reasoning, and the crucial novelty and effectiveness of the Cognitive Consistency Verification (CCV) module.We have carefully considered all points and provide detailed responses to the weaknesses (W1) and questions (Q1) below.
>
> ---
> >**W1: Missing video agent baseline: there were already some papers adopted a similar high-level idea of selecting video segments/frames for long video reasoning in a coarse-to-fine manner [1]. Therefore, adding at least one video agent baseline can make the contribution stronger.**
>
> A1: Thanks for your concern. Our method is grounded in the model's **internal** reasoning, allowing the model to determine how to explore and when to terminate. In contrast, methods like VCA require multiple invocations of **external** models in each round for observation and decision-making. Therefore, the main contribution of paper lies in pioneering an end-to-end learnable RL framework for this active reasoning paradigm instead the coarse-to-fine video processing mannner.
>
>
> Moreover, we incorporate three video agent methods into the table below for a comprehensive comparison on the challenging LVBench:
>
> | Method              | Base Model    | Frame |  Acc | End-to-End |
> | :------------------ |  :-----: | :----: | :-----: | :-----: |
> | VideoAgent (ECCV24)         | GPT-4o       | 25.5         | 29.3        | ✗          |
> | VideoTree (CVPR25)          | GPT-4o        | 103.2        | 28.8        | ✗          |
> | VCA (ICCV25)                | GPT-4o     | 20.0         | 41.3        | ✗      |
> | FrameThinker (Ours) | Qwen2.5-VL-7B | 23.9         | 36.6        | ✓      |
>
> The results present that our FrameThinker significantly surpasses VideoAgent and VideoTree based on GPT-4o, despite using the open-source Qwen2.5-VL-7B as its foundation. Due to more complex workflow design than previous works, VCA achievs better performance than our method. However, such ability is still rely on strong GPT-4o and requires two GPT-4o calls in each turn for observation and decision-making, while our model can decide appropriate action in one call. Therefore, although based on Qwen2.5-VL-7B, our method can achieve superior  efficiency and performance without relying on powerful closed-source models or fixed workflow.
>
>
>
> [1] Videoagent: Long-form video understanding with large language model as agent, in ECCV 2024
>
> [2] Videotree: Adaptive tree-based video representation for llm reasoning on long
> video, in CVPR 2025
>
> [3] Video Curious Agent for Long Video Understanding, in ICCV 2025

---

> > ### Comment · Reviewer_Qf4G · 2025-11-20
> >
> > Thanks for the comparisons! It makes the results stronger. I raised my rating accordingly.

---

> ### Author Response · Authors · 2025-11-20
> **Response to Reviewer Qf4G (part2)**
>
> >**Q1: Benchmarks in the paper are all long-video understanding and reasoning. Will training the model (i.e., FrameThinker) negatively impact the short-video reasoning, and traditional tasks for vision language models?**
>
> A1: Thanks for your question. FrameThinker does not suffer from performance degradation on short-video and traditional tasks.
>
> To illustrate, we evaluate FrameThinker on two short-video reasoning benchmark (MMVU, VideoTT), and two traditional tasks, including temporal grounding (CharadesSTA), and anomaly detection (UCF-Crime). The results are shown as below:
>
> For Short-Video QA (MMVU, VideoTT):
> |               |  |    MMVU (51s)  |  |     VideoTT (27s) |
> | :------------ | :----------: | :----------: | :----------: | :----------: |
> |               | Frame      | Acc  | Frame         | Acc  |
> | Video-R1      | 32         | 64.2 | 32            | 36.6 |
> | VideoChat-R1  | 32         | 60.6 | 32            | 34.2 |
> | Qwen2.5-VL-7B | 32         | 56.2 | 32            | 28.2 |
> | FrameThinker  | 16.1       | 64.8 | 16.0          | 37.1 |
> | ∆             | -50%       | +8.6 | -50%          | +8.9 |
>
> For Temporal Grounding (CharadesSTA):
> |               | Frame | mIoU | R@0.3 | R@0.5 | R@0.7 |
> | ------------- | :---: | :---: | :---: | :---: | :---: |
> | Video-R1      | 32    | 26.1 | 39.2  | 26.4  | 13.4  |
> | Qwen2.5-VL-7B | 32    | 28.2 | 42.4  | 28.2  | 13.5  |
> | FrameThinker  | 32    | 30.3 | 45.3  | 30.5  | 15.1  |
> | ∆             |       | +2.1 | +2.9  | +2.3  | +1.6  |
>
> For Anomaly Detection (UCF-Crime):
> |               | Frame | Acc  |Precise  |Recall  |F1-Score  |
> | ------------- | :---: | :---: |:---: |:---: |:---: |
> | Video-R1      | 32    | 55.5 |100 |7.9 |0.15 |
> | Qwen2.5-VL-7B | 32    | 71.7 |92.9 |47.1 |0.63 |
> | FrameThinker  | 16.5  | 76.9 |86.9 |62.8 |0.73 |
> | ∆             | -48%  | +5.2 | -6.0|+15.7 |+0.10 |
>
> From the results, we can obtain that:
> 1. FrameThinker significantly improves accuracy on both MMVU and VideoTT (by +8.6% and +8.9% respectively) while using 50% fewer frames than the baseline, demonstrating that its strategic reasoning is efficient and beneficial even for short videos.
> 2. For temporal grounding task, FrameThinker achieves a notable improvement on CharadesSTA. The enhancement is attributed to the learned ability to select and reason about time segments, which elevates its core grounding capability.
> 3. FrameThinker outperforms the baseline and other RL models on Anomaly Detection task UCF-Crime, suggesting that the RL training enhances the model's ability to focus on critical visual segments (Frame Spotlighting) for accurate event detection.

---

> > ### Comment · Reviewer_Qf4G · 2025-11-20
> >
> > Thanks for the added experiments. I have raised my rating.

---

> > > ### Author Response · Authors · 2025-11-21
> > > **Response to Reviewer Qf4G**
> > >
> > > Thank you for your positive feedback. We sincerely appreciate your time and recognition.

---

### Official Review · Reviewer_U4yG · 2025-11-01

**Soundness:** 3
**Presentation:** 3
**Contribution:** 4
**Rating:** 6
**Confidence:** 4

**Summary:**

This paper explores the task of long-video reasoning in Large Vision-Language Models (LVLMs). The author’s key motivation is that most existing LVLMs process frames uniformly sampled at a fixed interval, which often leads to many irrelevant frames being processed. To address this, they propose FrameThinker to enable LVLMs to perform frame sampling through a learned reasoning process. FrameThinker is trained in two stages: Supervised Fine Tuning (SFT) to teach structured thought-action generation, and Reinforcement Learning (RL) with GRPO to refine capabilities learned in the SFT stage. They conduct an extensive study on the RL stage, showing that unconditional and format-based rewards cause training collapse, and propose the Cognitive Consistency Verification (CCV) module to enforce alignment between thoughts and actions to stabilize the RL training.

**Strengths:**

The problem is well motivated as uniform sampling quickly becomes impractical as video tasks increase in complexity. As complexity increases, LVLMs will require the ability to reason about which frames to process rather than relying on dense or uniform sampling

FrameThinker is sufficiently different from existing frame sampling methods the reviewer is aware of, which rely on heuristics/pre-trained models or decouple training of the frame sampler and LVLM. In contrast, FrameThinker directly optimizes the LVLM to perform frame sampling

The method is presented clearly though the writing, and its effectiveness is shown on well-chosen benchmarks

**Weaknesses:**

Concern with comparisons to baseline model
* It seems like the baseline model (Qwen2.5-VL-7B) is evaluated zero-shot on the Video-Holmes and LongVideo-Reason datasets, but FrameThinker is first fine-tuned on these two datasets. The reviewer believes a true fair baseline in Table 1 would be Qwen2.5-VL fine-tuned on the Video-Holmes dataset, and In Table 2 and Table 3 it should be the Qwen2.5-VL fine-tuned on the SFT+RL instruction pairs without the thinking and action reasoning

FrameThinker still relies on uniform sampling in its initial spare scan and uses them to decide where to “zoom in” and focus. Given that a max of 12 frames are sampled, it is not unlikely that all of these frames will be irrelevant in some cases, especially for queries like the one in Figure 11 that are not contextualized with temporal information. In these cases, FrameThinker will suffer from the same limitation of uniform sampling as existing LVLMs

Minor comments on formatting:
* Line 26: “FrameThinker get” should be “FrameThinker gets”
* Notation: In Section 3.1 the query is defined as $i$ but in Section 3.2 the query is defined as $q$ (the $i$ also becomes a subscript instead of a superscript of Tau in Section 3.2). It seems to me like $i$ should correspond to a specific trajectory rollout and not the input query
* Suggestion for Table 2 and Table 3: It might be better if the deltas are green instead of red as Red makes the differences appear negative. The best performing models can also be bolded (consistent with Table 1)

**Questions:**

Can the authors clarify the evaluation protocol of the Qwen2.5-VL baseline (see Weakness 1)? If it is zero-shot, is there a reason why Qwen2.5-VL cannot be fine-tuned under the same conditions as FrameThinker?

How does FrameThinker handle cases where uniformly sampled frames dont capture any query-relevant frames (see Weakness 2)? I imagine it would randomly select a frame interval on which to “zoom in”, does FrameThinker have the ability to “zoom out”?

---

> ### Author Response · Authors · 2025-11-20
> **Response to Reviewer U4yG**
>
> We sincerely thank the reviewer for recognizing the strong motivation of addressing the impracticality of uniform sampling, and the difference of FrameThinker's approach in directly optimizing the LVLM to perform the frame selection policy. We have carefully considered all points and provide detailed responses to the weaknesses (W1-W3) and questions (Q1-Q2) below.
>
> ---
> >**W1 & Q1: Can the authors clarify the evaluation protocol of the Qwen2.5-VL baseline (see Weakness 1)? If it is zero-shot, is there a reason why Qwen2.5-VL cannot be fine-tuned under the same conditions as FrameThinker?**
>
> A1: Thanks for your concern. Our evaluation protocol of the Qwen2.5-VL in Table 1, 2, 3 follow the standard reporting convention of recent prominent works (e.g., Video-R1, VideoChat-R1, LongVILA-R1), where the performance of the base model is reported in a zero-shot manner for contextual comparison.
>
> In Table 1, we have compared our FrameThinker with GRPO-CARE, which is  a variant that fine-tuned Qwen2.5-VL on the Video-Holmes. The results show that FrameThinker (56.1) significantly surpasses GRPO-CARE (40.7). In Table 2, we also compare against GRPO-CARE* (trained on Video-Holmes) and LongVILA-R1 (trained on LongVideo-Reason), which are both trained with SFT + RL.
>
> Moreover, it is expensive to train models in large-scale datasets. Therefore, we do not study the variant of SFT+RL in Table 2 and Table 3. Instead, we have explored the variants of SFT and RL on Video-Holmes, as shown in Figure 4(a) of the paper. The results demonstrate that our FrameThinker can effectively learn thinking and action reasoning from these two stages.
>
>
> ---
> >**W2 & Q2: How does FrameThinker handle cases where uniformly sampled frames dont capture any query-relevant frames? I imagine it would randomly select a frame interval on which to “zoom in”, does FrameThinker have the ability to “zoom out”?**
>
>
>
> A2: Thanks for your question. When the initial sparse sampling can not capture information directly relevant to the question in the first turn, the model still gains a broad understanding of the video's overall theme and structure. This initial global context is indispensable for guiding the subsequent Multi-Turn Exploration steps. As demonstrated in Figure 11, the initial sparse frames set the stage for the search. This capability is further reinforced by our design that the full history, including the initial frames, is used as context for all subsequent turns, allowing the model to make informed decisions even after a failed first attempt.
>
> Notably, FrameThinker has the equivalent of a "zoom out" capability. Our framework retains the entire historical trajectory as context for every subsequent turn. This means that after performing a choose frames action (a "zoom in"), the model can effectively "zoom out" by generating a choose frames action with a significantly wider temporal range that encompasses the failed segment or even a completely new segment far from the previous focus. This is the Multi-Interval Exploration strategy described in **Appendix G.2, E**, where the model reverts to its initial, sparse global view to inform a new search. We have added a new **Figure 12 in Appendix D** (marked in blue in the paper) to further demonstrate this ability.
>
>
> ---
> >**W3: Minor Comments on Formatting**
>
> A3: We sincerely thank the reviewer for your careful reading and for pointing out these formatting issues. We will universally use q to represent the input query and use i as the index for a sampled trajectory, revise the tables and fix the mistake. We have updated these contents in the revised version. (marked in blue in the paper)

---

> > ### Author Response · Authors · 2025-11-27
> > **Looking forward to your reply!**
> >
> > Dear Reviewer U4yG,
> >
> > We hope this message finds you well. We are writing to kindly follow up on our response to your valuable comments and questions.
> >
> > In our rebuttal, we addressed your comments by clarifying the evaluation protocol for our baseline. We also explained how FrameThinker handles scenarios with uninformative initial frames, detailing its "zoom out" capability and adding a new example (`see Appendix D, Figure 12`). Additionally, we have corrected the formatting issues you pointed out in the revised manuscript (marked in blue in the paper).
> >
> > We believe that we have clarified the concerns and questions raised. If you have any additional thoughts or questions about our responses, we would be more than happy to address them.
> >
> > Thank you once again for your insightful review! We look forward to your further feedback.
> >
> > Best,
> >
> > The authors

---

> ### Comment · Reviewer_U4yG · 2025-11-28
>
> Dear authors,
>
> Thank you for the response and reminder!
>
> **Regarding W1 and Q1**
>
> Thank you for your response, but I may have been unclear in my initial review. What I want to know is: given the same amount of data, will pure SFT outperform SFT + RL? In Figure 4a, my understanding was that `Qwen (SFT)` is a VLM trained only on the SFT data presented in Section 3.2, while `Qwen (SFT+RL)` is initialized from Qwen (SFT) but is tuned on *additional* data from Video-Holmes. Is this correct or is this a misunderstanding, and both models are trained on the same amount of data?
>
> **Regarding W2 and Q2**
>
> Thank you for these details and the additional Figure. This ability of FrameThinker to “zoom out” is noted and is interesting, it definitely strengthens the usefulness of FrameThinker. I will raise my rating to 8 after the clarification on W1 and Q1.

---

> > ### Author Response · Authors · 2025-11-28
> > **Response to Reviewer U4yG**
> >
> > Thanks for your positive response. About Figure 4a, your understanding is correct. Qwen (SFT+RL) is initialized from Qwen (SFT) and is tuned on additional data from Video-Holmes. This experiment verifies that the RL stage could effectively improve performance atop the SFT model.
> >
> > To compare the ability of pure SFT and SFT + RL under the same amount of data, we randomly sampled 1196 examples from both the SFT and RL datasets to ensure the total data count matches that of the SFT baseline (2392).  We then performed SFT followed by RL using this data subset. The results are shown below:
> > |  | SFT (2392)   | SFT (1196) + RL (1196)  |
> > | :------------------ | :---: | :---: |
> > | **Accuracy**  | 35.9  | 39.9  |
> >
> > This result further demonstrates the superiority of SFT+RL. Compared to the imitation learning in SFT, RL better stimulates the model's reasoning capabilities.

---

### Official Review · Reviewer_wput · 2025-11-05

**Soundness:** 3
**Presentation:** 3
**Contribution:** 3
**Rating:** 6
**Confidence:** 4

**Summary:**

This paper addresses long video reasoning by introducing FrameThinker, a framework that enables a vision-language model to iteratively select and analyze frames across multiple reasoning turns. The model begins with a coarse video scan and selectively retrieves frames for closer inspection based on its evolving reasoning. It is trained in two stages: supervised fine-tuning to learn tool syntax, followed by reinforcement learning to optimize its frame selection policy. Key contributions include new action primitives, a multi-turn reasoning paradigm, and carefully designed reward mechanisms for stable training. The method achieves state-of-the-art accuracy on video QA benchmarks while using far fewer frames—for instance, 76.1% on LongVideo-Reason with ~20 frames versus 72.0% from prior work with 512. Overall, FrameThinker delivers significant accuracy gains (+10.4% on average) with substantially higher efficiency.

**Strengths:**

- Novelty - The paper introduces a new multi-turn “thinking” paradigm for video understanding, which is notable in this context, although not a fundamentally new algorithm. By allowing the model to iteratively query the video (via learned actions) rather than passively reading a fixed set of frames, it bridges the gap between static video QA models and interactive video agent systems.
- Methodology - The two-phase training—supervised fine-tuning followed by RL—is well designed, with thoughtful reward structuring to prevent pitfalls like mode collapse. The Cognitive Consistency Verification module adds robustness, and ablation studies confirm that each component, including multi-turn reasoning and CCV, contributes to performance. The use of GRPO and carefully tuned rewards further support the method’s effectiveness.
- Empirical Performance - The paper shows substantial improvements in both accuracy and efficiency across multiple long video reasoning benchmarks. FrameThinker outperforms all baselines while using significantly fewer frames—for example, achieving 76.1% on LongVideo-Reason with ~20 frames versus 72.0% with 512. This efficiency makes it well-suited for scaling to longer videos.
- Impact - This work addresses a key bottleneck in video AI – the inability to efficiently handle long videos by enabling models to focus on relevant frames, offering a scalable solution for long-horizon video analysis. Its integration of reinforcement learning into vision-language models introduces a flexible inference strategy with potential impact beyond video.

**Weaknesses:**

- Related Work - This section should have an explicit section on video key frame selection/sampling, given the focus of the paper. Quite a few key papers are missing from the discussion [a,b,c,d]
- Frame Selection Ablation - One potential concern is the lack of comparison to other frame selection strategies. The paper convincingly shows improvements over uniform sampling and static baselines, but we don’t see comparisons to any heuristic or learned frame selection method [a,b,c,d]
- Efficiency - The paper does not report actual inference time or compute cost comparisons. It’s assumed fewer frames = faster, but the iterative process might introduce some overhead (multiple forward passes). Quantifying the real-time speedup (or trade-off) would strengthen the empirical claims of efficiency.
- Evaluation Scope - The experiments, while extensive on the benchmarks provided, focus mainly on QA tasks. It’s not fully explored how the approach would perform on other types of long video understanding tasks
- Failure Cases - The paper would benefit from a clearer discussion of failure cases, such as missed events due to poor initial scans or premature stopping. It’s unclear how often issues like over-exploration occur or whether the Cognitive Consistency Verification (CCV) module mistakenly blocks valid strategies. Providing insight into these patterns and how frequently CCV intervenes would help assess the method’s robustness and reliability.

References.
- [a] M-LLM Based Video Frame Selection for Efficient Video Understanding, CVPR 2025
- [b] VideoEspresso: A Large-Scale Chain-of-Thought Dataset for Fine-Grained Video Reasoning via Core Frame Selection, CVPR 2025
- [c] Vila: Efficient videolanguage alignment for video question answering, ECCV 2024
- [d]  Self-Chained Image-Language Model for Video Localization and Question Answering, NeurIPS 2023

**Questions:**

- Frame Selection Ablation - Have you considered comparing FrameThinker to heuristic or learned frame selection strategies beyond uniform sampling? Including such baselines would help clarify how much of the performance gain comes from the learned policy versus the general benefits of dynamic frame selection.
- Speedup - What is the computational speedup from processing fewer frames? The paper shows a drastic reduction in frames (e.g., 20 vs 512), but due to the multi-turn approach, there may be multiple forward passes. How does the actual inference time or FLOPs compare to a single-pass baseline? Some discussion or measurement of runtime efficiency would strengthen the claim of “unparalleled efficiency.”
- Generality - How adaptable is FrameThinker beyond QA tasks? Could it handle video captioning or anomaly detection without an explicit query, or is a well-defined question essential for guiding frame selection? Exploring this would strengthen the paper.

**Details Of Ethics Concerns:**

Not Applicable.

---

> ### Author Response · Authors · 2025-11-20
> **Response to Reviewer wput (part1)**
>
> We sincerely thank the reviewer for recognizing the novelty of our multi-turn “thinking” paradigm, the thoughtful design of the two-phase training and CCV module, and our substantial empirical performance. We have carefully considered all points and provide detailed responses to the weaknesses (W1-W5) and questions (Q1-Q3) below.
>
> ---
> >**W1: Related Work - This section should have an explicit section on video key frame selection/sampling.**
>
> A1: Thanks for your suggestion. We have added a discussion on key frame selection in section **related work** (marked in blue in the paper) as follows:
>
> **Video Key Frame Selection**. Recent advancements have intensely focused on efficient key frame selection to combat the long video bottleneck. Methods like SEVILA, ViLA, M-LLM Based Video Frame Selection, and VideoEspresso all employ mechanisms, such as localizers or question-aware selectors, to score or pick relevant frames from an initial sampled set, drastically reducing visual tokens for downstream reasoning. FrameThinker, however, adopts a fundamentally different paradigm: rather than pre-processing a sampled collection of frames, it uses multi-turn reasoning to directly output temporal interval boundaries for frame sampling, which makes FrameThinker end-to-end learnable from data.
>
> ---
> >**W2 & Q1: Frame Selection Ablation - One potential concern is the lack of comparison to other frame selection strategies. The paper convincingly shows improvements over uniform sampling and static baselines, but we don’t see comparisons to any heuristic or learned frame selection method [a,b,c,d]**
>
> A2: Thanks for your insightful review. Following your suggestion, we conduct an additional comparison with paper [c] VILA and [d] SEVILA. The results are shown as below:
> |              |         | NExTQA |       | STAR |
> | :----------- |  :-----: | :----: | :-----: | :-----:
> |              | Frame   | Acc    | Frame | Acc  |
> | SEVILA [d]      | 32 to 4 | 63.6   | 32 to 4 | 44.6 |
> | VILA [c]        | 32 to 4   | 74.4   | --     | --    |
> | VILA [c]        | --  |--       |4    | 67.1|
> | FrameThinker | 15.8    | 75.9   | 16.0  | 72.7 |
>
> As shown in this table, our FrameThinker surpasses previous work [c] and [d], achieving superior performance with only a small number of frames.
>
>
>
> ---
> >**W3: Efficiency - The paper does not report actual inference time or compute cost comparisons.
> >Q2: Speedup - How does the actual inference time or FLOPs compare to a single-pass baseline?”**
>
> A3: Thanks for your review. To demonstrate the real-world efficiency of FrameThinker, We conduct an evaluation below focusing on Wall-Clock Latency (Time) on LongVideo-Reason benchmark.
>
> | Model                   | Frame    | Acc      | Time (s) |
> | :---------------------- | :-------: | :-------: | :-------: |
> | Video-R1                | 32       | 68.1     | 8.6      |
> | LongVILA-R1             | 256      | 70.8     | 38.1     |
> | LongVILA-R1             | 512      | 72.0     | OOM      |
> | **FrameThinker (Ours)** | **20.6** | **76.1** | **7.7**  |
>
> From the results, we can obtain:
>
> - At 256 frames, LongVILA-R1 is nearly 5 times slower than FrameThinker, and the 512-frame setting (official result reported) causes OOM in our testing, underscoring the severe computational cost of high frame counts.
> - Even at an  32 frame count, Video-R1 is slower than FrameThinker (8.6s vs 7.7s) because it is designed to evoke deep, single-shot reasoning that results in a longer Chain-of-Thought generation time.
>
> The comparison demonstrates that FrameThinker achieves superior performance not only in terms of frame efficiency but also in end-to-end Wall-Clock Latency.

---

> ### Author Response · Authors · 2025-11-20
> **Response to Reviewer wput (part2)**
>
> >**W4: Evaluation Scope - It’s not fully explored how the approach would perform on other types of long video understanding tasks.
> > Q3: Generality - How adaptable is FrameThinker beyond QA tasks? Could it handle video captioning or anomaly detection without an explicit query, or is a well-defined question essential for guiding frame selection? Exploring this would strengthen the paper.**
>
> A4: Thanks for your concern. Actually, a well-defined question is essential for guiding the model's strategic, multi-turn frame selection. However, this **does not limit** FrameThinker to traditional tasks if these tasks can be framed as a query for our model. For instance, we can translate the Anomaly Detection task with below prompt:
>   >Does the following video contain any of the anomalous behaviors: Abuse, Arrest, Arson, Assault, Burglary, Explosion, Fighting, RoadAccidents, Robbery, Shooting, Shoplifting, Stealing, Vandalism? A: Yes, B: No.
>
>
> To verify the effectivenes of this strategy, we perform experiments on UCF-Crime dataset of Anomaly Detection. The results show that FrameThinker significantly outperforms the baseline and Video-R1. Notably, FrameThinker achieves much **higher recall**, suggesting our multi-turn frame spotlighting is especially effective at locating anomalies in the video.
>
> |               | Frame | Acc  |Precise  |Recall  |F1-Score  |
> | ------------- |  :-----: | :----: | :-----: | :-----: |:-----: |
> | Video-R1      | 32    | 55.5 |100 |7.9 |0.15 |
> | Qwen2.5-VL-7B | 32    | 71.7 |92.9 |47.1 |0.63 |
> | FrameThinker  | 16.5  | 76.9 |86.9 |62.8 |0.73 |
> | ∆             | -48%  | +5.2 | -6.0|+15.7 |+0.10 |
>
>
>
> While FrameThinker could be prompted to caption the video, organizing its multi-turn thought process to write a single caption is challenging. The specific outcome highly depends on the precise prompt or query chosen for such tasks.
>
> ---
> >**W5: Failure Cases - The paper would benefit from a clearer discussion of failure cases, such as missed events due to poor initial scans or premature stopping. It’s unclear how often issues like over-exploration occur or whether the Cognitive Consistency Verification (CCV) module mistakenly blocks valid strategies.**
>
> A5: Thanks for your suggestion. We have provided detailed examples of critical failure modes in appendix, **E. Bad Case**.
>
>
>  For failure cases of poor initial scans or premature stopping, we added an extra section in **Appendix E.4** (marked in blue in the paper) discussing a failure case when poor initial scans failed to provide the model with sufficient information and the model subsequently committed premature stopping in the later process. Regarding the concern of over-exploration, we investigated the relationship between the number of exploration rounds and performance in **Section 3.3.4**, along with the corresponding failure cases in **Appendix E.2**. By utilizing multi-round rewards, we observed that more rounds do not necessarily lead to better results. Consequently, our final implementation is designed to discourage excessive exploration. Therefore, over-exploration is not a significant issue during practical inference.
>
>
> Moreover, the CCV module will not mistakenly block valid strategies. This is because CCV is strictly designed as a rule-based filter that only intervenes for absolute, fundamental logical inconsistencies (e.g., repeating the same action, thought-action contradiction). It does not evaluate the quality of the strategic path but only ensures the basic rationality of the thought-action sequence. Therefore, a valid, complex strategy cannot be incorrectly flagged by the CCV.

---

> > ### Author Response · Authors · 2025-11-27
> > **Looking forward to your reply!**
> >
> > Dear Reviewer wput,
> >
> > We hope this message finds you well. We are writing to kindly follow up on our response to your valuable comments and questions.
> >
> > In our rebuttal, we addressed your comments by expanding the related work discussion (`see Section 2`), adding a comparison with other frame selection methods, conducting a new experiment on wall-clock latency (`see Section 4.5`), performing an experiment on a non-QA task (anomaly detection), and including an analysis of failure cases (`see Appendix E.4`).
> >
> > We believe that we have clarified the concerns and questions raised. If you have any additional thoughts or questions about our responses, we would be more than happy to address them.
> >
> > Thank you once again for your insightful review! We look forward to your further feedback.
> >
> > Best,
> >
> > The authors

---

### Official Review · Reviewer_gjov · 2025-11-06

**Soundness:** 3
**Presentation:** 3
**Contribution:** 2
**Rating:** 4
**Confidence:** 3

**Summary:**

This paper proposes a LVLM that actively reasons over long videos by alternating thoughts and actions in multiple turns, trained via SFT then RL with a rule-based Cognitive Consistency Verification (CCV) filter to keep thoughts and actions logically aligned. The SFT process is trained to learn observe-think-action process, and the RL is to learn what are the frames and when in the timestamp to sample. The integrated CCV is to surpass the illogical thought-action pairs. Results on LongVideo-Reason, Video-Holmes demonstrate the method's frame efficiency. However, there are some drawbacks such as reward shaping is fragile (unconditional or naive bonuses can cause collapse) and CCV is rule-based rather than learned, so behavior may hinge on handcrafted checks and hyperparameters

**Strengths:**

1. Prior LVLMs, Qwen2.5-VL, LongVILA-R1, mostly do one-shot reasoning on a big uniformly sampled frame set. FrameThinker instead actively selects frames over multiple turns, so it hits higher accuracy with far fewer frames.
2. Compared to Video-R1 and VideoChat-R1, which are also built upon Qwen2.5-VL-7B, FrameThinker consistently scores higher on long-video benchmarks while using similar or fewer frames, showing that the active frame reasoning policy actually adds value beyond the backbone.
3. Video agents, e.g., VideoAgent, RIVET-like systems, typically rely on manually designed workflows or external tools and are not trained end-to-end; they follow a pre-defined pipeline for querying frames or detectors. While this method makes frame operations part of the LVLM’s own action space and trains them with RL, so the policy for "where to look next" is learned from data, not hard-coded. That’s a big step up in autonomy and adaptability vs those agent-style baselines.
4. Compared to many video agents, FrameThinker adds Cognitive Consistency Verification (CCV) that checks redundancy and fidelity of thought-action pairs, so the proposed method can more easily spot and filter illogical exploration instead of trusting a black-box planner.

**Weaknesses:**

1. The paper’s efficiency claims are measured only in terms of the number of frames processed before feeding to the fixed LVLM, which is not an end-to-end computation. Compared to prior RLVR and CoT baselines that perform a single-shot pass on a uniformly sampled frame set, FrameThinker introduces extra multi-turn reasoning with repeated LVLM calls over successively updated contexts. While this is likely more frame-efficient, it is unclear whether it is actually more computationally efficient overall.
2. The method is not compared against any SOTA that also performs reasoning-based or active frame sampling on long videos, making it difficult to justify the benefits of the proposed design.
3. Can the method be plug-and-play module before any LVLMs? The proposed FrameThinker requires an action grammar, a bespoke RL setup, a rule-based CCV module, and nontrivial reward tuning.

**Questions:**

1. The authors measure efficiency mainly as "frames processed per question." How does FrameThinker compare to LongVILA-R1 / Video-R1 in terms of actual FLOPs or wall-clock latency, given you do multiple LVLM passes per example?
2. Why not compare against a baseline that uses uniform frames but also multi-turn CoT (repeated LVLM calls) to match your compute pattern more fairly?
3. How would FrameThinker adapt to long-video tasks where the supervision is not QA-style (e.g., temporal localization, dense captioning, or video editing assistance) where rewards and clean correctness signals are harder?
4. How many examples does the model answer correctly without really using new frames (i.e., from the question or initial sparse scan alone)? Do you detect significant "shortcut" behavior?
5. How robust are results to the exact CCV rules and thresholds? If you relax or slightly perturb them, does performance drop sharply?

---

> ### Author Response · Authors · 2025-11-20
> **Response to Reviewer gjov (part1)**
>
> We sincerely thank the reviewer for recognizing FrameThinker's paradigm shift to active reasoning that achieves higher efficiency and accuracy, its value beyond the backbone, and its advancement in autonomy and adaptability over agent baselines through the novel CCV module. We have carefully considered all points and provide detailed responses to the weaknesses (W1-W3) and questions (Q1-Q5) below.
>
> ---
> >**W1 & Q1: The paper’s efficiency claims are measured only in terms of the number of frames processed. How does FrameThinker compare to LongVILA-R1 / Video-R1 in terms of actual FLOPs or wall-clock latency?**
>
> A1: Thanks for your concern. We conduct an evaluation below focusing on Wall-Clock Latency (Time) on LongVideo-Reason.
>
> | Model                   | Frame    | Accuracy      | Wall-Clock Latency (s) |
> | :---------------------- | :-------: | :-------: | :---------------------: |
> | Video-R1                | 32       | 68.1     | 8.6                    |
> | LongVILA-R1             | 256      | 70.8     | 38.1                   |
> | LongVILA-R1             | 512      | 72.0     | OOM                    |
> | **FrameThinker (Ours)** | **20.6** | **76.1** | **7.7**                |
>
> From the table, we can obtain that:
> - At 256 frames, LongVILA-R1 is nearly 5 times slower than FrameThinker, and the 512-frame setting (official result reported) causes OOM in our testing, underscoring the severe computational cost of high frame counts.
> - Even at an  32 frame count, Video-R1 is slower than FrameThinker (8.6s vs 7.7s) because it is designed to evoke deep, single-shot reasoning that results in a longer Chain-of-Thought generation time.
>
> The comparison clearly demonstrates that FrameThinker achieves superior performance not only in terms of frame efficiency but also in end-to-end Wall-Clock Latency.
>
>
> ---
> >**W2: The method is not compared against any SOTA that also performs reasoning-based or active frame sampling on long videos.
> >Q2: Why not compare against a baseline that uses uniform frames but also multi-turn CoT (repeated LVLM calls) to match your compute pattern more fairly?**
>
> A2: Thanks for your question. In our paper, we have already compared with **reasoning-based methods**, such as Video-R1, VideoChat-R1, GRPO-CARE, LongVila-R1, which also leverage reinforcement learning to stimulate the model's reasoning capabilities. Table 2 in section 4 presents that FrameThinker achieves higher accuracy while using fewer frames.
>
> Moreover, we conduct an additional comparison against Video Agent methods that employ **active frame sampling**, under the challenging LVBench as below. This comparison clearly demonstrates our superior performance. It also verifies the benefits of our end-to-end learnable framework, namely the strategic policy for visual exploration and decision-making is optimized directly by the model.
>
> | Method                  | Base Model        | Frame | Acc | End-to-End |
> | :---------------------- | :----------------: | :-----------: | :----------: | :---------: |
> | VideoAgent [1] (ECCV24)             | GPT-4o            | 25.5         | 29.3        | ✗          |
> | VideoTree [2] (CVPR25)             | GPT-4o            | 103.2        | 28.8        | ✗          |
> | **FrameThinker (Ours)** | **Qwen2.5-VL-7B** | **23.9**     | **36.6**    | **✓**      |
>
>
> In addition, as requested in Q2, we directly compare against a **baseline that uses uniform frames but also multi-turn CoT** to match our compute pattern more fairly. Specifically, for this baseline, we perform two-turn reasoning. In the first turn, we input 32 frames and ask the model to output a chain-of-thought. Then, in the second turn, we use the initial input and the thought from the first turn to ask the model to verify its answer. The results on LVBench are as follows:
>
> |               | Frame | Acc  |
> | ------------- | :-----: | :----: |
> |Qwen2.5-VL-7B（Single）      | 32    | 31.6 |
> | Qwen2.5-VL-7B（Multi） | 32    | 31.1 |
> | **FrameThinker (Ours)**  | 23.9  | 36.6 |
>
> We found that direct multi-turn reasoning actually reduced performance. We attribute this to the approach's lack of a mechanism for dynamically updating the visual context, which causes the static multi-turn Chain-of-Thought to accumulate errors and dilute context. FrameThinker, in contrast, uses an active, and iterative frame spotlighting mechanism to ensure the model can dynamically retrieve and focus on new key visual evidence.
>
>
> [1] Videoagent: Long-form video understanding with large language model as agent, In ECCV 2024
> [2] Videotree: Adaptive tree-based video representation for llm reasoning on long video, In CVPR 2025

---

> ### Author Response · Authors · 2025-11-20
> **Response to Reviewer gjov (part2)**
>
> >**W3: Can the method be plug-and-play module before any LVLMs? The proposed FrameThinker requires an action grammar, a bespoke RL setup, a rule-based CCV module, and nontrivial reward tuning.**
>
> A3: Thanks for your review. Currently, FrameThinker relies on training and is not plug-and-play. However, **each component** in our FrameThinker is highly reusable and flexibly transfer to any LVLMs:
>
>   1. Action Grammar: The structured output format (<think>…</think><action>…</action>) and the action grammar itself (e.g., choose frames, get frame number) are highly reusable. This grammar defines a universal interface for iterative visual exploration that can be applied to any LVLM.
>   2. RL Setup: The two-stage training paradigm (SFT to instill basic grammar, followed by RL to learn strategic policy) coupled with Group Relative Policy Optimization (GRPO) is a highly reusable recipe for LVLM.
>   3. Rule-Based CCV Module: Its simple checks can be adopted by other RL-based LVLMs to ensure the generated thought-action trajectories remain logically grounded.
>   4. Reward Tuning: Our in-depth analysis of the RL reward provides reusable, general insights for the design of other LVLMs. These discoveries offer a guiding principle for robustly designing reward functions.
>
>
> ---
> >**Q3: How would FrameThinker adapt to long-video tasks where the supervision is not QA-style (e.g., temporal localization, dense captioning, or video editing assistance) where rewards and clean correctness signals are harder?**
>
> A4: Thanks for your insightful question. Actually, the complexity of obtaining clean correctness signals varies significantly across tasks.
>
> For other non QA-style video tasks, such as temporal localization, it is feasible to organize the problem into a question-answering format using an appropriate prompt, directly asking the model to output the time interval based on the query.
>
> To verify the effectiveness, we conduct Temporal Grounding on Charades-STA datasets and the results are shown as below. FrameThinker also achieves a notable improvement on Charades-STA. The imporvement is attributed to the learned ability to select and reason about time segments, which elevates its core grounding capability. Note that currently we are only performing inference, without specific training. If training is required, IoU can be further used as a reward.
>
> |               | Frame | mIoU | R@0.3 | R@0.5 | R@0.7 |
> | ------------- | :-----: | :----: | :-----: | :-----: |:-----: |
> | Video-R1      | 32    | 26.1 | 39.2  | 26.4  | 13.4  |
> | Qwen2.5-VL-7B | 32    | 28.2 | 42.4  | 28.2  | 13.5  |
> | FrameThinker  | 32    | 30.3 | 45.3  | 30.5  | 15.1  |
> | ∆             |       | +2.1 | +2.9  | +2.3  | +1.6  |
>
> For more subjective tasks like dense captioning or video editing assistance, obtaining a clean, binary reward is challenging. To solve this problem, recent works [1][2] use LLM as Judge or scorer to obtain a scalar reward. Our framework can likewise obtain the necessary reward and train for these tasks.
>
> [1] VideoCap-R1: Enhancing MLLMs for Video Captioning via Structured Thinking
>
> [2] VersaVid-R1: A Versatile Video Understanding and Reasoning Model from Question Answering to Captioning Tasks

---

> ### Author Response · Authors · 2025-11-20
> **Response to Reviewer gjov (part3)**
>
> >**Q4: How many examples does the model answer correctly without really using new frames (i.e., from the question or initial sparse scan alone)? Do you detect significant "shortcut" behavior?**
>
> A5: We appreciate this insightful question and conduct a two-part analysis to address it:
>
> 1. To verify that the model genuinely uses new frames when it requests them, we isolate all trajectories that are answered correctly and involved at least one choose frames action. We then employ an LLM-as-a-Judge protocol using Qwen2.5-VL-7B to classify the reasoning path for each trajectory with the following prompt:
>
> >Based on this thought chain, which of the following best describes how the model arrived at the correct answer?
> A. The model answered correctly using new frames.
> B. The model answered correctly from the question alone.
> C. The model answered correctly from the initial sparse scan alone.
>
>   The analysis revealed that 100% of these multi-turn reasoning paths were classified as Category A. This strongly indicates that when FrameThinker decides to take an action to retrieve new frames, it genuinely utilizes the new visual evidence to reach its final conclusion, effectively mitigating detrimental "shortcut" behavior.
>
> 2. Moreover, for some simple questions, our model does answer correctly based on the initial sparse scan alone. However, this is not a shortcut. In such cases (e.g., **Figure 13 in Appendix D**), the model correctly identifies that the initial evidence is sufficient and proceeds directly to the output answer action in a single turn. This behavior is optimal as it avoids multi-turn overhead and further reduces frame consumption, contributing to the model's overall high efficiency.
>
> In summary, FrameThinker effectively avoids "shortcut" behavior by utilizing new frames when requested, while also demonstrating strategic efficiency by forgoing action when the initial context is sufficient.
>
> ---
> >**Q5: How robust are results to the exact CCV rules and thresholds? If you relax or slightly perturb them, does performance drop sharply?**
>
> A5: Thanks for your question. CCV module is a strict, rule-based binary logical filter without thresholds. It is designed to return either 1 (pass) or 0 (fail) based on absolute logical correctness. Moreover, Figure 4.c shows that even without the CCV module, the performance does not degrade significantly. This is because our main innovation and performance improvement lie in the multi-turn frame spotlighting. The primary role of the CCV is to ensure **logical consistency** and enhance **interpretability**.

---

> > ### Author Response · Authors · 2025-11-27
> > **Looking forward to your reply!**
> >
> > Dear Reviewer gjov,
> >
> > We hope this message finds you well. We are writing to kindly follow up on our response to your valuable comments and questions.
> >
> > In our rebuttal, we addressed your comments by conducting new experiments to compare wall-clock latency with baselines (`see Section 4.5`), adding further comparisons against other active sampling methods, and clarifying the flexible and transferable nature of our framework's components. We also included an experiment on a non-QA task, provided an analysis demonstrating that our model genuinely utilizes new frames, and explained the robustness of the CCV module.
> >
> > We believe that we have clarified the concerns and questions raised. If you have any additional thoughts or questions about our responses, we would be more than happy to address them.
> >
> > Thank you once again for your insightful review! We look forward to your further feedback.
> >
> > Best,
> >
> > The authors

---

### Official Review · Reviewer_jZYr · 2025-11-06

**Soundness:** 4
**Presentation:** 4
**Contribution:** 4
**Rating:** 6
**Confidence:** 3

**Summary:**

The paper introduces a novel framework, FrameThinker, that reframes long video understanding. Other than using a traditional single-pass processing, FrameThinker uses multi-turn frame reasoning process, which dynamically highlights the relevant frames. By generating textual thoughts hint guided by actions prior, it focus on promising segments by choosing specific frame ranges.

In specific, it didn’t load large set of frames but learn to engages in a multi-turn reasoning loop. Then it utilizes the dynamic actions strategies which the primary action is chosen with the start and end frame within the “highlighted” range, allowing to extract the critical information based on its reasoning. After that, they uses a two-phase pipeline for training: 1. Using a supervised fine-tuning tiny dataset as teaching model for the syntax and mechanics of actions; 2. Using a larger dataset with reinforcement learning to train a strategic policy on controlling the timing to use these actions for. They provides a comprehensive reward design correspondingly for this task in order to point out unconditional rewards that lead to collapse and set conditional action bonus only on final success. In addition, they introduce a cognitive consistency verification (CCV) module to suppose illogical executions and ensure the interpretability of the model’s actions.

The author demonstrates the performance of FrameThinker by conducting extensive experiments to gain superior accuracy while using significantly fewer frames.

**Strengths:**

•	The paper tackles a critical challenge in long video understanding: the efficiency and interpretability of long video due to uniform frame sampling for long-term reasoning. The motivation is interesting, and the core idea of “multi-turn frame spotlighting”. The author successfully builds up a pipeline of combining thoughts and actions along with observations. It mimics human-like analysis (skimming, then focusing) and moves the field from passive processing to active, agentic reasoning.

•	Cognitive Consistency Verification (CCV): The paper carefully addresses the reward collapse issue during training with an unconditional action bonus. The CCV module is a novel and effective build-up component that acts as a filter after the rollout process in order to validate every trajectory by checking possible redundancy, logical flow, and fidelity. CCV ensures the generated chain of thoughts during the decision-making process.

•	SOTA on extensive experiments: With fewer input frames as final feed, delta and performance are consistent. FrameThinker reaches SOTA performance on xis benchmarks(Video-Holmes, LongVideo-Reason, LongVideoBench, MLVU, VideoMME-Long, LVBench).

**Weaknesses:**

•	Action Space Flexibility: The paper mentioned that the current action space will depend on the selected range. It can help pinpoint the action to make predictions more accurate, but will also limit the generalizability of the model if it fails to get enough action within the range.

•	CCV’s robustness: Since CCV is a rule-based module, it weakens its generalizability when facing a more complex, interactive scenario, which creates a more difficult reasoning path. CCV might lose its advantage in complex scenes.

**Questions:**

•	Would the CCV flag a valid but complex reasoning path as an error, and how do you plan to scale these rules to more complex reasoning?

•	During the SFT phase (Phrase 1 training), if the teacher failed to give a basic action syntax for the features of SFT, it might likely to significantly decrease the performance on the final performance since it mainly relies on the quality of the teacher model’s data.

---

> ### Author Response · Authors · 2025-11-20
> **Response to Reviewer jZYr (part1)**
>
> We sincerely thank the reviewer for recognizing our framework's strong motivation on long video efficiency, the novelty of CCV, and achieving SOTA performance across six benchmarks with significantly fewer frames. We have carefully considered all points and provide detailed responses to the weaknesses (W1-W2) and questions (Q1-Q2) below.
>
> ---
> >**W1: Action Space Flexibility: The paper mentioned that the current action space will depend on the selected range. It can help pinpoint the action to make predictions more accurate, but will also limit the generalizability of the model if it fails to get enough action within the range.**
>
>
> A1: Thanks for your concern. The flexible and context-rich design of FrameThinker ensures that its generalizability is not limited. The reasons are twofold:
>
> 1. Global Context Preservation: The current action space **does not solely rely on the range selected in the previous turn**, but rather is conditioned on the **entire historical trajectory**. This comprehensive context includes all interleaved initial sparse frames, the full reasoning text, and all frames retrieved from previous steps (as conceptually illustrated by the leftmost arrows in **Figure 2(a)**). This design allows our model to maintain a view of the global content while focusing locally. Consequently, even if a local "zoom-in" segment fails to yield effective information, the model can still rely on the global context to continue its reasoning.
> 2. Explicit Training for Iterative Exploration: The current design has already incorporated the scenario where a locally focused search is unsuccessful and the model needs to "re-think" and jump to another local segment. This capability is explicitly trained using the **E. Multi-Interval Exploration** template within the process-supervised data of the SFT phase (**Appendix G.2, Line 1440**). We have added an example in **Appendix D, Figure 12** to further demonstrate. (Mark in blue in the paper.)
> ---
> >**W2: CCV’s robustness: Since CCV is a rule-based module, it weakens its generalizability when facing a more complex, interactive scenario, which creates a more difficult reasoning path. CCV might lose its advantage in complex scenes.**
>
> A2: Thanks for your review. CCV module will not lose its advantage in complex scenarios. It only penalizes trajectories that can be absolutely and unequivocally judged as incorrect. For example, the Redundancy Check flags any instance where the exact same action with identical parameters is executed more than once. This check is still effective in complex scenes.
>
> Moreover, the CCV module remains critical in complex scenarios for preventing **fundamental** logical inconsistencies, which is vital for the stability and rationality of the entire multi-turn process. The simplicity and objectivity of rule-based CCV module makes it a robust, low-overhead component, essential for ensuring the logical consistency of complex multi-turn reasoning.

---

> > ### Author Response · Authors · 2025-11-20
> > **Response to Reviewer jZYr (Part 2)**
> >
> > >**Q1: Would the CCV flag a valid but complex reasoning path as an error, and how do you plan to scale these rules to more complex reasoning?**
> >
> > A1: Thanks for your concern. The CCV module will not flag a valid but complex reasoning path as an error. As elaborated in above answer to W2, this module is meticulously designed to target only absolute and **fundamental** logical inconsistencies.
> >
> > Moreover, CCV is easy to scale to more complex reasoning. The core rules are defined in a way that **naturally accommodates a greater number of turns**. For example, the Redundancy Check and the Logical Flow Check are inherently designed to operate over the entire, arbitrarily long trajectory. Therefore, the current set of rules can scale effortlessly to multi-turn scenarios, providing essential logical supervision without inhibiting the model's capacity for complex generalization.
> >
> > ---
> > >**Q2: During the SFT phase, if the teacher failed to give a basic action syntax for the features of SFT, it might likely to significantly decrease the performance on the final performance since it mainly relies on the quality of the teacher model’s data.**
> >
> > A2: Thanks for your question. It is **easy to learn** basic action syntax in the SFT phrase and the quality of the teacher model’s data is not a bottleneck. As summarized in the table below, we measure the format accuracy when training with different amounts of SFT data:
> >
> > | SFT Data (Examples) | 0    | 500  | 1000 | +RL |
> > | :------------------ | :---: | :---: | :---: | :-------: |
> > | **Correct Format**  | 90.2%  | 92.9%  | 97.4%  | 100%     |
> >
> > The results clearly show:
> > - Even with zero SFT data (relying only on the instruction prompt), the model achieves a respectable 90.2% format accuracy, demonstrating the strong inherent capability of the base model (Qwen2.5-VL-7B) to follow new syntax.
> > - The use of a small SFT dataset (500-1000 examples) is highly effective, quickly pushing the format accuracy to 97.4%.
> > - The subsequent RL phase eliminates all remaining format errors, leading to 100% correct format accuracy in the final model.
> >
> > These confirm that the SFT phase efficiently instills the required action syntax and **the subsequent RL stage can further imporve this ability**. Therefore, the risk of performance degradation due to teacher-model syntax failure in SFT is minimal.

---

> > > ### Author Response · Authors · 2025-11-27
> > > **Looking forward to your reply!**
> > >
> > > Dear Reviewer jZYr,
> > >
> > > We hope this message finds you well. We are writing to kindly follow up on our response to your valuable comments and questions.
> > >
> > > In our rebuttal, we addressed your comments by explaining the multi-turn mechanism and adding an example to illustrate it (`see Appendix D, Figure 12`). We also clarified the robustness and scalability of the CCV module, and included an experiment to demonstrate that the SFT phase can effectively learn the required action syntax.
> > >
> > > We believe that we have clarified the concerns and questions raised. If you have any additional thoughts or questions about our responses, we would be more than happy to address them.
> > >
> > > Thank you once again for your insightful review! We look forward to your further feedback.
> > >
> > > Best,
> > >
> > > The authors

---

### Meta-Review · Area_Chair_eq9G · 2026-01-07

**Summary:**

The submission introduces a long-form video understanding framework that enables a vision-language model to iteratively select and analyze frames across multiple reasoning turns. The main contribution is on the reward design for GRPO learning, where the merit of the proposed CCV module is supported by extensive evaluations.

The submission received initial ratings of 66644, after rebuttal reviewer U4yG promised to raise the score from 6 to 8 if the training data usage was clarified (the authors provided additional evaluation with the same amount of data, which to the AC's understanding addresses the concern); reviewer Qf4G found the additional rebuttal experiments addressed their concerns on missing baselines (which the AC agrees with), and commented to raise their rating (presumbly from 4 to 6). For reviewer gjov, the concerns were alternative efficiency metrics (frame-level and overall computation), missing comparison with SoTA methods (e.g., multi-turn CoT), and the ease of use of the proposed method as an "plug-and-play" module. The AC believes the first questions have been addressed by the rebuttal, while the last question is a minor limitation. Overall, the AC agrees with the reviewers' assessments, believes that the rebuttal has successfully addressed all major concerns, and therefore should be accepted by ICLR.

**Reviewer Concerns:**

As discussed above, concerns from reviewers U4yG, gjov, Qf4G have been successfully addressed by the rebuttal.

For reviewer jZYr, there were minor concerns on action space flexibility and CCV robustness, which the AC believes the rebuttal has addressed both.

For reviewer wput, there were concerns on related work, efficiency, ablation, and failure cases. The AC believes the rebuttal and revision have addressed the concerns.

**Reviewer Scores:**

The AC believes that it's likely that reviewer U4yG will upgrade the rating from 6 to 8, and Qf4G will upgrade the rating from 4 to 6. It is unclear whether reviewer gjov will update their rating, but as discussed the AC believes most of their concerns have been addressed.

---

### Decision · Program_Chairs · 2026-01-26

Accept (Poster)